# Super-resolution mapping in rod photoreceptors identifies rhodopsin trafficking through the inner segment plasma membrane as an essential subcellular pathway

Kristen N. Haggerty[1]☯, Shannon C. Eshelman [1]☯, Lauren A. Sexton[1], Emmanuel Frimpong[1], Leah M. Rogers[1], Melina A. Agosto [2], Michael A. Robichaux [1]*

1 Department of Ophthalmology & Visual Sciences and Department of Biochemistry & Molecular Medicine, West Virginia University, Morgantown, West Virginia, United States of America, 2 Retina and Optic Nerve Research Laboratory, Department of Physiology and Biophysics, and Department of Ophthalmology and Visual Sciences, Dalhousie University, Halifax, Nova Scotia, Canada

☯ These authors contributed equally to this work.
* michael.robichaux@hsc.wvu.edu

**Data Availability Statement:** All confocal microscopy image data are available in the published article. SIM raw data and STORM

## Abstract

Photoreceptor cells in the vertebrate retina have a highly compartmentalized morphology for efficient phototransduction and vision. Rhodopsin, the visual pigment in rod photoreceptors, is densely packaged into the rod outer segment sensory cilium and continuously renewed through essential synthesis and trafficking pathways housed in the rod inner segment. Despite the importance of this region for rod health and maintenance, the subcellular organization of rhodopsin and its trafficking regulators in the mammalian rod inner segment remain undefined. We used super-resolution fluorescence microscopy with optimized retinal immunolabeling techniques to perform a single molecule localization analysis of rhodopsin in the inner segments of mouse rods. We found that a significant fraction of rhodopsin molecules was localized at the plasma membrane, at the surface, in an even distribution along the entire length of the inner segment, where markers of transport vesicles also colocalized. Thus, our results collectively establish a model of rhodopsin trafficking through the inner segment plasma membrane as an essential subcellular pathway in mouse rod photoreceptors.

## Introduction

In the vertebrate retina, vision is initiated by the phototransduction cascade in the outer segment (OS) sensory cilia of rod and cone photoreceptor cells. The light-absorbing G-protein coupled receptor (GPCR) proteins rhodopsin (Rho) in rods and the cone opsins in cones are densely packaged into flattened membrane discs that are stacked within the OS cilium. In the mouse retina, each rod OS contains approximately 800 discs [1] with a Rho packaging density

molecule list data are openly available via Mendeley Data at DOI:10.17632/pv3ty3z68f.1, DOI: 10.17632/r4246hbxsw.1, and DOI: 10.17632/hmztjrwhtc.1. Original western blot scans are available in source data files.

**Funding:** This work was supported by the National Institute of Health (https://www.nih.gov/): P20-GM144230 (MAR); Knights Templar Eye Foundation to MAR and Discovery Grant (RGPIN-2022-02982) from the Natural Sciences and Engineering Research Council of Canada to MAA. The funders had no role in study design, data collection and analysis, decision to publish, or preparation of the manuscript.

**Competing interests:** The authors have declared that no competing interests exist.

**Abbreviations:** BB, basal body; BSA, bovine serum albumin; CC, connecting cilium; co-IP, coimmunoprecipitation; DAP, distal appendage; ER, endoplasmic reticulum; FWHM, full width half maximum; GAP, GTPase activating protein; GC1, guanylate cyclase 1; GMII, Golgi alpha-mannosidase II; GPCR, G-protein coupled receptor; IFT, intraflagellar transport; immuno-EM, immunoelectron microscopy; IS, inner segment; LCA, Leber congenital amaurosis; NbGFP-A647, GFP nanobody Alexa 647 conjugate; NGS, normal goat serum; OCT, optimal cutting temperature; ONL, outer nuclear layer; OPL, outer plexiform layer; OS, outer segment; PDC, phosducin; PSF, point spread function; RAIN-STORM, rapid imaging of tissues at the nanoscale STORM; Rho, rhodopsin; RP, retinitis pigmentosa; SIM, structured illumination microscopy; SNARE, soluble N-ethylmaleimide-sensitive factor attachment protein receptor; STORM, stochastic optical reconstruction microscopy; STX3, syntaxin 3; TEM, transmission electron microscopy; WGA, wheat germ agglutinin.

of approximately 75,000 Rho molecules per disc [2–4]. Rod OS discs are completely renewed in about 10 days through the process of disc shedding from the distal rod OS tips [5,6]. As such, new discs are formed at the OS base, and these nascent discs must be filled with newly synthesized protein, primarily Rho, which is trafficked through the rod photoreceptor inner segment (IS) to the OS by way of the connecting cilium (CC) [7,8]. In mouse rods, the CC is a thin, approximately 300-nm-diameter ciliary bridge that spans 1.1 μm between the IS and OS and is composed of an axoneme core of 9 microtubule doublets that extend into the OS [9]. A coordinated homeostasis of OS disc shedding, nascent disc formation, IS protein delivery, and CC trafficking must be maintained in photoreceptors for a lifetime of proper vision.

Mislocalization of Rho is a hallmark phenotype reported in many animal models of retinal disease, including models for retinitis pigmentosa (RP) and other retinal ciliopathies. RP is an inherited neurodegeneration affecting 1:4,000 in the United States of America [10] that causes neuronal cell death in rod photoreceptors and subsequent loss of vision. In animal models, a wide range of Rho RP mutations have been demonstrated to cause Rho protein mislocalization to the rod IS region, as well as some degree of Rho mislocalization to the photoreceptor outer nuclear layer (ONL) and outer plexiform layer (OPL) [11–15]. RP mutations to other rod-specific genes also result in Rho mislocalization [16–19], and mouse models for syndromic retinal ciliopathies and Leber congenital amaurosis (LCA) are caused by mutations in cilia-related genes that lead to rod degeneration in the retina due to defective ciliary ultrastructure that is comorbid with Rho mislocalization [20–24]. Collectively, rod dystrophies caused by any number of disruptions to normal rod homeostasis lead to Rho mislocalization, such that restorative therapeutic strategies to treat these retinal diseases must account for proper trafficking of Rho to preserve long-term rod stability. Therefore, a thorough understanding of the cellular mechanisms of Rho synthesis and trafficking in the IS of rods is essential for the development of effective retinal therapies.

The rod IS compartment is the biosynthetic domain of rods that is filled with endoplasmic reticulum (ER) and Golgi secretory organelles, mitochondria, and cytoplasmic microtubules all surrounded by a plasma membrane. The rod IS microtubular network is proposed to nucleate from the basal body (BB) [8,25], which is a region of the apical IS composed of 2 centrioles, one of which—the mother centriole—is continuous with the axoneme of the CC. Cytoplasmic dynein 1 has been demonstrated to be essential for rod health as the putative motor protein complex for intracytoplasmic movement toward the minus end of microtubules in the IS [26–29]. The BB is also the site of the distal appendages (DAPs), which are 9 radially symmetrical pinwheel-shaped blades that link the mother centriole to the plasma membrane and may serve as a gate or barrier structure at the critical IS/CC interface (reviewed in Wensel and colleagues' study [30]). The mouse rod IS also contains a ciliary rootlet, which is a filamentous cytoskeletal element that is linked to the BB and extends to the rod presynaptic terminal [31]. Still, there is uncertainty regarding the subcellular localization and integration of Rho and other nascent proteins within the IS compartment architecture.

In the IS of frog rod photoreceptors, Rho has been localized with immunoelectron microscopy (immuno-EM) with the Golgi, in post-Golgi Rho carrier vesicles, and sporadically at the IS plasma membrane [32–34]. In early immuno-EM mammalian retina studies, Rho was localized to the IS plasma membrane in both mouse and cow rods [35], and in immunolabeled thin sections of pig retina, Rho was grossly localized to the IS plasma membrane, Golgi, and ONL [36]. More recent images of post-embedding immuno-EM or cryo-immuno-EM localization of Rho in adult mouse rods have shown minimal Rho labeling in the IS [26,37,38–42]. Despite the lack of comprehensive Rho IS localization details from these studies, Rho has been consistently localized at mouse CC plasma membrane in mouse rods [38–40,42]. In fluorescence localization studies using mammalian retina, including in hRho-GFP fusion knock-in mouse

retinas, it has to date been challenging to develop methodology to visualize the small population of Rho molecules in the IS compared to the overwhelming amount of Rho in OS discs [43].

The network of small GTPase protein regulators of the Rho secretory pathway in the IS has been systematically defined in frog rods. In this pathway, Arf4 GTPase binds to Rho in the *trans*-Golgi network via the Rho C-terminal VxPx motif [44], a sequence that was shown to be essential for OS targeting in frogs and zebrafish [45,46]. ASAP1, the Arf GTPase activating protein (GAP), and small GTPases FIP3 and Rab11a are then recruited to the complex to form post-Golgi Rho carrier vesicles [47,48], which are then targeted to the plasma membrane by Rab8a GTPase and Rabin8, the Rab8 guanine nucleotide exchange factor [49–51]. In mice, Arf4, Rab8, and Rab11a are not essential for Rho trafficking [52,53], suggesting the existence of redundant regulators or compensatory trafficking networks in mouse rods. One common Rho trafficking mechanism is the role of IS SNARE (soluble N-ethylmaleimide-sensitive factor attachment protein receptor) proteins. Rho vesicle docking to the IS plasma membrane in frog rods requires the proteins syntaxin 3 (STX3) and SNAP25 [54]. In the retinas from rod-specific STX3 knockout mice, Rho and the OS disc rim protein peripherin-2 are mislocalized to the IS and ONL [55], demonstrating that the functional role for STX3 is maintained in mice. Furthermore, STX3 was also shown to form in vivo protein interactions with Rho [56]. After SNARE-mediated docking, Rho-containing vesicles putatively fuse with the plasma membrane and Rho protein gets inserted into the membrane, an event that is classically mapped near the BB and CC in the apical IS of frog rods [33], but this key event has not been thoroughly examined in the mouse rod IS.

Overall, considering the current uncertainty regarding the impact of protein regulators of Rho biosynthesis and trafficking in mice, the complete route of Rho trafficking within the IS remains incompletely characterized despite the importance of trafficking within this critical subcellular compartment in mouse rods. In this study, we used super-resolution fluorescence microscopy to perform a quantitative spatial analysis of Rho localization in mouse rods ISs. Super-resolution modalities are powerful tools for testing the subcellular localization of protein targets in mouse rod domains, such as the IS. Both structured illumination microscopy (SIM), and stochastic optical reconstruction microscopy (STORM), a single-molecule localization microscopy, were previously used to localize ciliary proteins to the nanometer-scale subcompartments of the CC in mouse rods [24,57]. More recently, an alternative 3D STORM mode, named rapid imaging of tissues at the nanoscale STORM (RAIN-STORM), was developed to localize proteins in rod photoreceptor presynaptic terminals and postsynaptic dendrites within the mouse OPL [58]. To enable super-resolution localization of Rho in the mouse IS, we applied techniques to reliably immunolabel Rho in the IS of mouse retina, including an OS peeling approach, nanobody targeting of Rho-GFP in Rho-GFP/+ knock-in retinas, and surface labeling for SIM and STORM followed by a quantitative localization analysis.

## Results

### Rhodopsin immunofluorescence labeling strategies in mouse retina

Previous studies established that immunolabeling the N-terminus or C-terminus of Rho labels different fractions of Rho molecules in rods from mammalian retinal tissue preparations [36,59]. The immunofluorescence staining pattern of Rho was tested using the following antibodies: (1) the 1D4 monoclonal antibody (hereafter Rho-C-1D4), which targets the last 8 amino acids in C-terminus/cytoplasmic tail of Rho, i.e., the 1D4 sequence [60]; (2) the N-terminus targeting 4D2 monoclonal antibody (Rho-N-4D2) [59]; and (3) an N-terminus targeting Rho-N-GTX polyclonal antibody. For immunofluorescence, the Rho antibodies were first

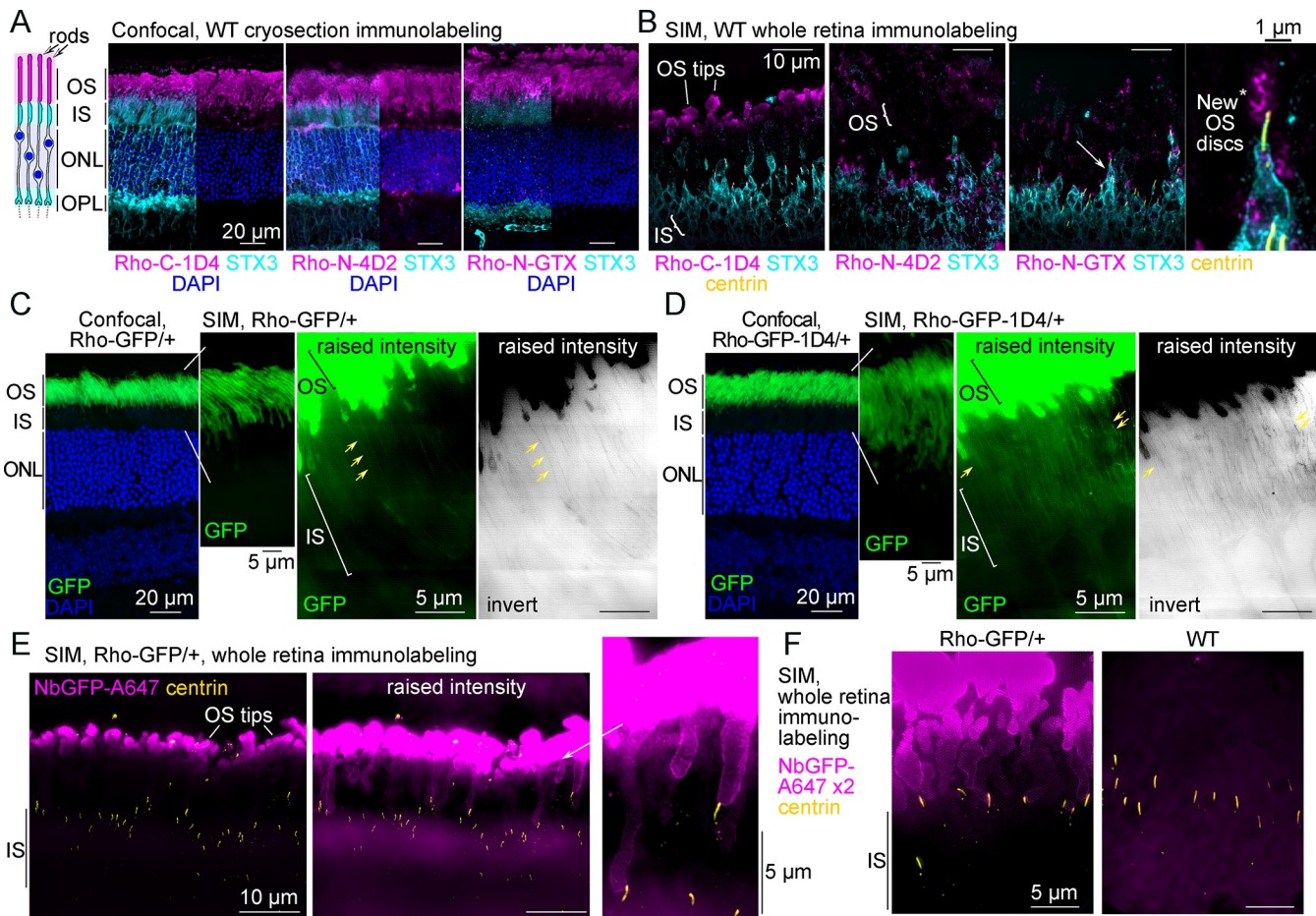

**Fig 1. Mouse rod inner segments are nonpermissive to conventional Rho labeling strategies.** (**A**) Diagram of mouse rod photoreceptor layers adjacent to z-projection confocal fluorescent images of WT mouse retinal cryosections immunolabeled for Rho (magenta), STX3 (cyan)—which labels the IS and OPL—and DAPI nuclear staining (blue) in the ONL. Rho prominently labels rod OS. The STX3 channel is removed from part of the images for clarity. (**B**) Z-projection SIM images of thin (1 μm) plastic sections of WT retinas stained for immunofluorescence as whole retinas. The white arrow indicates the rod that is magnified in the Rho-N-GTX example, and the white asterisk indicates Rho-N-GTX labeling of the new OS discs in the magnified rod. Centrin (yellow) immunolabeling was used to localize the rod CC and BB. (**C**) Z-projection confocal image of an adult Rho-GFP/+ cryosection adjacent to z-projection SIM images of 2 μm Rho-GFP/+ cryosections. (**D**) Z-projection confocal image of a Rho-GFP-1D4/+ cryosection and z-projection SIM images of 2 μm Rho-GFP/+ cryosections. In both (**C**) and (**D**), a magnified region of a SIM image is shown with raised contrast and brightness (intensity) levels to depict faint IS GFP fluorescence in both heterozygous knock-in mouse lines (yellow arrows). Inverted images are shown to highlight this pattern. (**E**) Z-projection SIM images of Rho-GFP/+ thin plastic retina sections that were immunolabeled for NbGFP-A647 (magenta) and centrin (yellow) as whole retinas. Raised intensity images are shown to depict less intense NbGFP-A647 labeling in some proximal OSs and surrounding the CC. (**F**) SIM images of thin sections from Rho-GFP/+ and WT retinas that were stained as in (**E**) but with twice the amount of NbGFP-A647 (x2). Lack of staining in WT retinas demonstrates NbGFP-A647 specificity. BB, basal body; CC, connecting cilium; IS, inner segment; ONL, outer nuclear layer; OPL, outer plexiform layer; OS, outer segment; Rho, rhodopsin; SIM, structured illumination microscopy; STX3, syntaxin 3; WT, wild-type.

tested in immunolabeled mouse retinal cryosections imaged with confocal scanning microscopy. With this method, all the Rho antibodies targeted the OS layer almost exclusively with strong fluorescence labeling (Fig 1A). Rho immunofluorescence localization in mouse retinas with SIM and STORM was also tested, which required a whole-retina immunolabeling approach modified from previous STORM imaging studies [24,57] for both improved labeling density and thin resin sectioning of stained retinas (see Materials and methods for details). By whole-retina labeling and SIM, the Rho-C-1D4 antibody only labeled the distal rod OS tips with fluorescence, while both N-terminal antibodies, Rho-N-4D2 and Rho-N-GTX, predominantly labeled the base of the OS where new discs are formed (Fig 1B). N-terminal Rho

immunolabeling at the OS base is consistent with the N-termini of Rho in newly formed evaginating discs being exposed to the extracellular space [61]. In each condition, Rho protein in the IS was not reliably detectable with immunofluorescence.

Next, 2 knock-in Rho fusion mouse lines were used: Rho-GFP and Rho-GFP-1D4. In each knock-in, GFP is fused to human Rho directly after the C-terminal 1D4 sequence [43]. In the Rho-GFP-1D4 version of the knock-in, an additional 1D4 sequence, which contains the VxPx targeting motif, is appended to the C-terminus of GFP [15]. Although both knock-in mice, as heterozygotes, have normal photoreceptor morphology and Rho OS localization, the addition of the extra 1D4 sequence was previously shown to be essential for GFP and Rho-Dendra2 OS targeting in frog and fish retinas [45,46,62], and so the Rho-GFP-1D4 knock-in mice was included in this mouse study to account for any possible subcellular localization requirement of the 1D4 sequence.

In confocal images of Rho-GFP/+ and Rho-GFP-1D4/+ retinas, GFP fluorescence was confined to in the OS layer, and only a faint, noise-like GFP signal was visualized in the IS layer in thin (2 µm) retinal cryosections from both knock-in mice using SIM (Fig 1C and 1D). Therefore, a GFP-specific nanobody was obtained, purified, and conjugated to Alexa 647, a bright and stable fluorophore compatible with both SIM and STORM, to both enhance the GFP signal in our knock-in mouse retinas and to overcome the issue of GFP bleaching during our whole retina processing technique. Nanobodies are small (approximately 15 kDa) single-chain camelid immunoglobulins that are ideal for super-resolution fluorescence microscopy [63]. The specificity of the GFP nanobody Alexa 647 conjugate (abbreviated NbGFP-A647) was validated with western blotting and confocal immunofluorescence (S1A–S1C Fig), in which WT retinas were used as controls for Rho-GFP labeling with NbGFP-A647. In each test, NbGFP-A647 properly labeled Rho-GFP or Rho-GFP-1D4 with no evidence of nonspecific labeling in WT controls. Notably, the Rho-C-1D4 antibody properly labeled the Rho-GFP-1D4 protein band in western blots of Rho-GFP-1D4/+ retinal lysates (S1B Fig). To validate these mouse lines further, we used a deglycosylation assay to confirm that Rho-GFP and Rho-GFP-1D4 fusion proteins, from Rho-GFP/+ and Rho-GFP-1D4/+ retina lysates, respectively, were properly glycosylated like the endogenous mouse Rho protein from the same lysates (S1D and S1E Fig).

Next, whole retina immunolabeling of Rho-GFP/+ and WT retinas with NbGFP-A647 was performed. In SIM images of thin sections from these Rho-GFP/+ retinas, NbGFP-A647 fluorescence was again predominantly limited to the distal OS tips, and only a few rods had any Rho-GFP labeling in the proximal OS near the centrin-positive CC (Fig 1E). Immunolabeling of centrins, which are calcium-binding proteins localized specifically to the CC and BB centrioles in mouse rods [57,64], was used to label the positions of the CC and BB as the apical edge of the IS throughout the study. Even when Rho-GFP/+ retinas were stained with double the amount of NbGFP-A647 (Fig 1F), the Rho-GFP-localized fluorescence was still predominantly found in the distal OS tips of Rho-GFP/+ retinas with no detectable fluorescent signal in the IS, indicating that NbGFP-A647 could not penetrate into the IS.

## Outer segment peeling of mouse retinas enables immunofluorescence detection of rhodopsin in the photoreceptor inner segment layer

To improve the penetration of our immunoreagents to the rod IS, a retinal peeling technique was used in which mouse retinas were iteratively adhered to and removed from filter paper to physically detach OSs from the retina in order to generate IS-enriched retina samples [65,66] (Fig 2A). OS removal using between 4 and 8 paper peeling cycles was tested using standard transmission electron microscopy (TEM). Eight peeling cycles resulted in increased OS

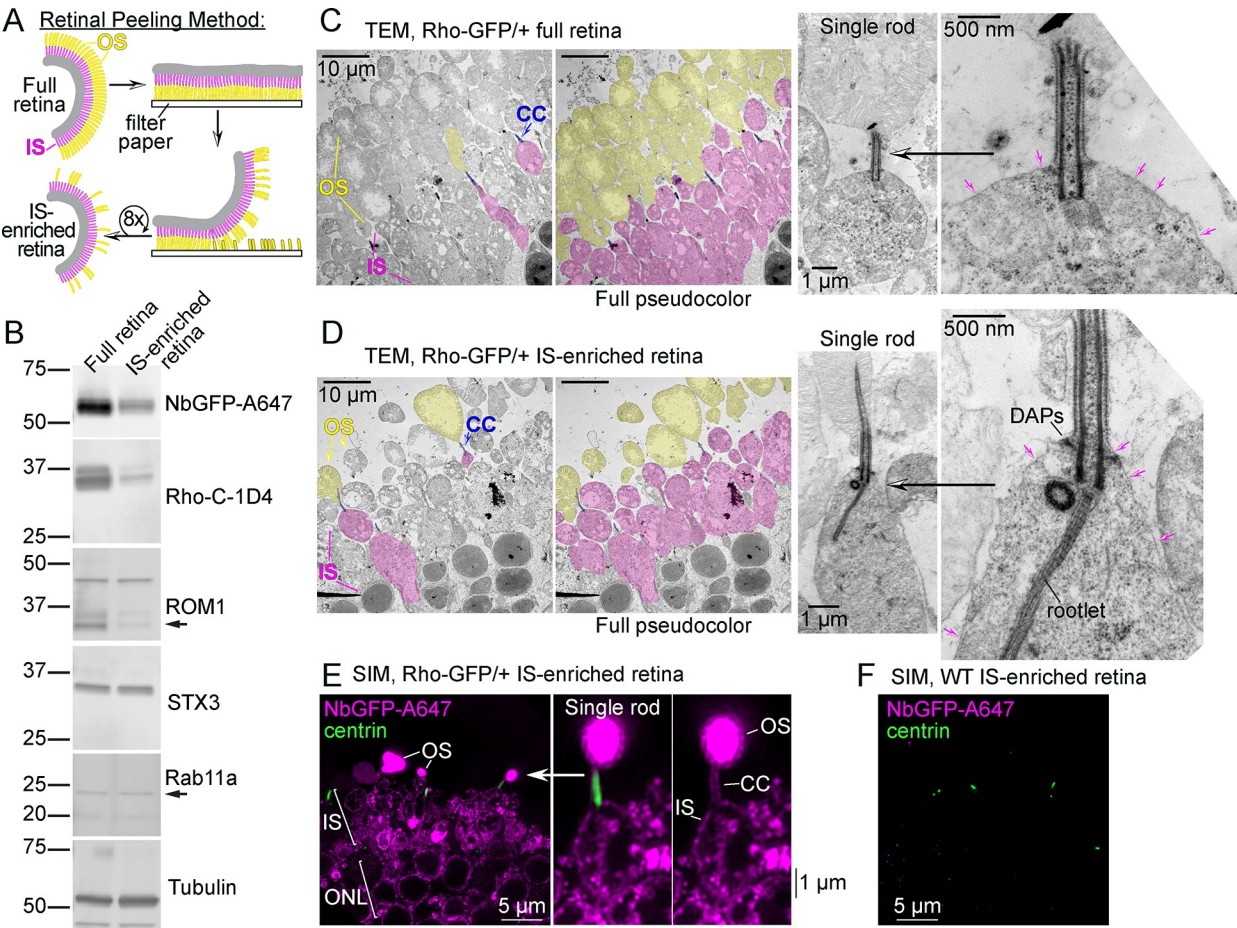

**Fig 2. OS peeling generates IS-enriched mouse retina samples.** (**A**) Diagram of the mouse retina OS peeling method. Mouse retina slices are iteratively peeled from filter paper 8 times to remove rod OSs (yellow) from the IS layer (magenta). (**B**) Western blot analysis comparing control full retina samples vs. retinas after OS peeling/IS-enriched retinas; both from Rho-GFP/+ mice. Approximately 2% of total volume from lysates from 1 retina (either control or peeled) were used from each condition for SDS-PAGE. Molecular weight marker sizes are indicated in kDa. Immunolabeled bands from antibodies targeting OS proteins, including NbGFP-A647, Rho-C-1D4, and ROM1 (black arrow points to the ROM1 monomer band), were reduced in peeled lysates, whereas IS proteins STX3 and Rab11a were roughly equal demonstrating IS enrichment. Tubulin immunoblotting was performed as a loading control. (**C**) Control full retina slices and (**D**) IS-enriched retinal slices from Rho-GFP/+ mice were fixed and stained for resin embedding, ultrathin sectioning, and TEM ultrastructure analysis. TEM images were pseudocolored to point out key rod structures as follows: OS = yellow, IS = magenta, CCs/BBs = blue. Single rod examples are magnified from both conditions to emphasize intact CC ultrastructure and the preservation of the IS plasma membrane (magenta arrows). The locations of the DAPs and rootlet are annotated in (**D**). (**E**) Z-projection SIM images of Rho-GFP/+ IS-enriched retinas immunolabeled with NbGFP-A647 (magenta) and centrin antibody (green). Fluorescence in the single rod example corresponding to the OS, IS, and CC are annotated. (**F**) Control SIM image of a WT IS-enriched retina immunolabeled and imaged as in (**E**) to demonstrate NbGFP-A647 labeling specificity. Scale bar values match adjacent panels when not labeled. BB, basal body; CC, connecting cilium; DAP, distal appendage; IS, inner segment; NbGFP-A647, GFP nanobody Alexa 647 conjugate; OS, outer segment; Rho, rhodopsin; STX3, syntaxin 3; SIM, structured illumination microscopy; TEM, transmission electron microscopy; WT, wild-type.

removal, and so this number was used throughout this investigation to generate IS-enriched retinas (S2A Fig). After generating IS-enriched retinas from Rho-GFP/+ mice, the enrichment of IS proteins was validated with western blotting, where the amount of OS proteins, Rho-GFP, endogenous mouse Rho (labeled with Rho-C-1D4 antibody), and ROM1, were reduced compared to full retina control samples, while the levels of IS proteins STX3 and Rab11a were unchanged between conditions (Fig 2B).

Next, Rho-GFP/+ retinas were fixed for TEM immediately after peeling/IS enrichment to evaluate if the IS and CC ultrastructure were maintained after OS removal. Compared to the

overall morphology of unpeeled Rho-GFP/+ rod photoreceptors (Fig 2C), IS-enriched retinas contained fewer OSs with a generally intact IS layer (Fig 2D). In higher-magnification TEM micrographs, IS-enriched rod examples had preserved CC structure that were typically attached to a remaining OS, and these rods had an intact IS plasma membrane and cytoplasm (Figs 2D and S2B). Because there was evidence that some ISs lacked CC as a result of the OS peeling, all IS-enriched retinas analyzed in this study were stained with CC/BB markers to ensure that only ISs that were not damaged by the peeling were analyzed.

IS-enriched retinas from Rho-GFP/+ mice were then immunolabeled with NbGFP-A647 using whole-retina immunolabeling and SIM as before. In this case, presumably due to the removal of many of the OSs in these retinas that would otherwise soak up all available immuno-reagents, NbGFP-A647 fluorescence was now observed in the rod CC, IS, and ONL regions among the sparse remaining OSs in these sections (Fig 2E). WT IS-enriched retinas were also stained with NbGFP-A647 to demonstrate labeling specificity (Fig 2F).

## Rhodopsin is localized at the mouse rod inner segment plasma membrane

Compared to confocal images of Rho-GFP/+ IS-enriched retinas co-immunolabeled with NbGFP-A647 and centrin antibody (Fig 3A), SIM clearly improved the signal-to-noise ratio of the immunofluorescence in our retina sections, which was further enhanced with 3D deconvolution (Fig 3B). In SIM images of single Rho-GFP/+ rods, Rho-GFP was prominently localized at the apparent boundary of the IS in a continuous string of fluorescence that was contiguous

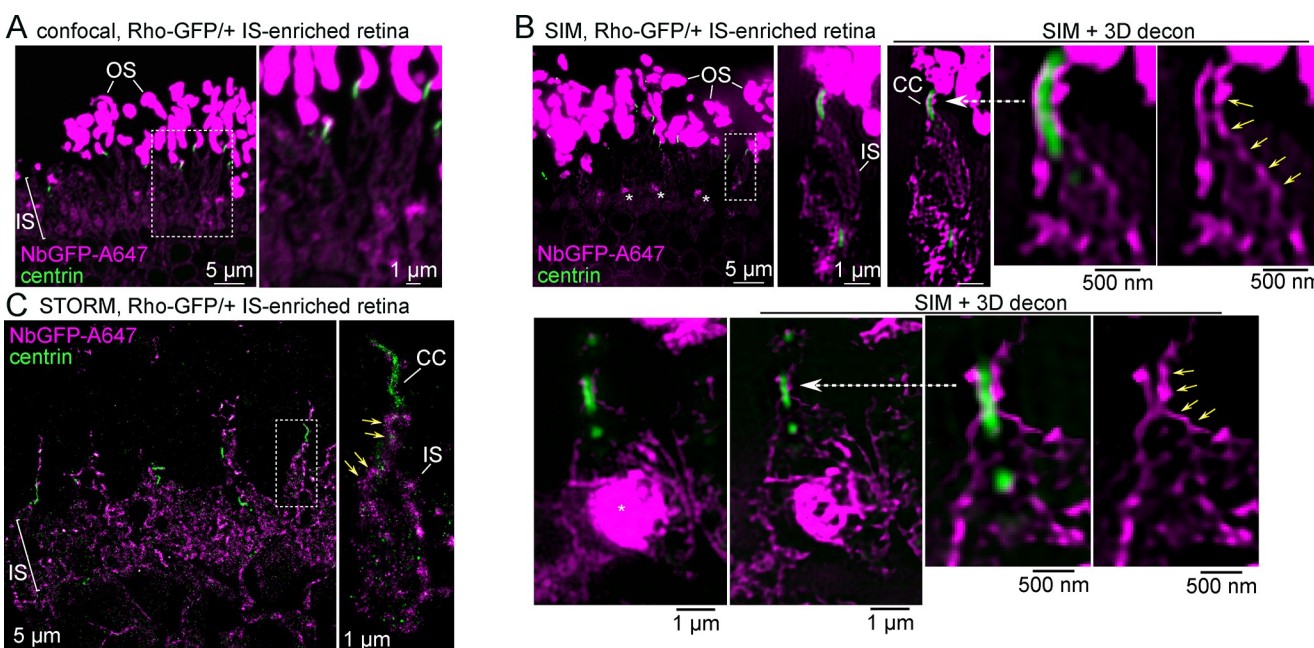

**Fig 3. A fraction of Rho-GFP molecules are localized at the rod IS plasma membrane.** (**A**) Confocal z-projection and (**B**) SIM z-projection images of an IS-enriched, Rho-GFP/+ retina section immunolabeled with NbGFP-A647 (magenta) and centrin antibody (green). Single rod examples are magnified in (**B**) to highlight Rho-GFP localization along the boundaries of the IS and CC. These same single rod images are also shown after 3D-deconvolution processing (SIM + 3D decon). Yellow arrows indicate a continuous line of Rho-GFP fluorescence between the IS and CC. Asterisks indicate strong Rho-GFP puncta staining in the proximal IS/myoid region. (**C**) STORM reconstruction of an IS-enriched Rho-GFP/+ retina section. In the magnified single rod examples, yellow arrows indicate Rho-GFP molecules located at the IS boundary. Magnified regions are indicated throughout with either a dashed box or a dashed white arrow. Some magnified images are rotated so that the OS end of the rod is at the top of the image. Scale bar values match adjacent panels when not labeled. CC, connecting cilium; IS, inner segment; NbGFP-A647, GFP nanobody Alexa 647 conjugate; OS, outer segment; Rho, rhodopsin; SIM, structured illumination microscopy; STORM, stochastic optical reconstruction microscopy.

with the CC plasma membrane (Fig 3B). Bright puncta of Rho-GFP were also observed in the proximal IS region, known as the myoid (Golgi-rich) IS (Fig 3B, asterisks); however, in single rod views, internal Rho-GFP in the distal/ellipsoid IS cell body was not reliably observed in any particular subcompartment. STORM single-molecule localization of NbGFP-A647 in Rho-GFP/+ IS-enriched retinas demonstrated the same localization pattern of Rho-GFP at the apparent rod IS plasma membrane (Fig 3C).

We sought a reliable antibody marker of the IS plasma membrane so that single ISs could be identified in our localization experiments. Antibodies that target STX3, a t-SNARE protein previously localized at the IS plasma membrane in mouse retina [55,67,68], and the Na/K-ATPase channel, which has also been used as a general IS marker [22,69], were tested. Using SIM and STORM super-resolution microscopy experiments, the STX3 antibody resulted in more densely labeled IS boundaries of individual rods (Fig 4A and 4B). Therefore, STX3 was used as the IS plasma membrane marker in subsequent experiments.

To test if Rho-GFP localizes at the rod IS plasma membrane, IS-enriched Rho-GFP/+ retinas were co-immunolabeled for NbGFP-A647 along with the STX3 antibody to mark the location

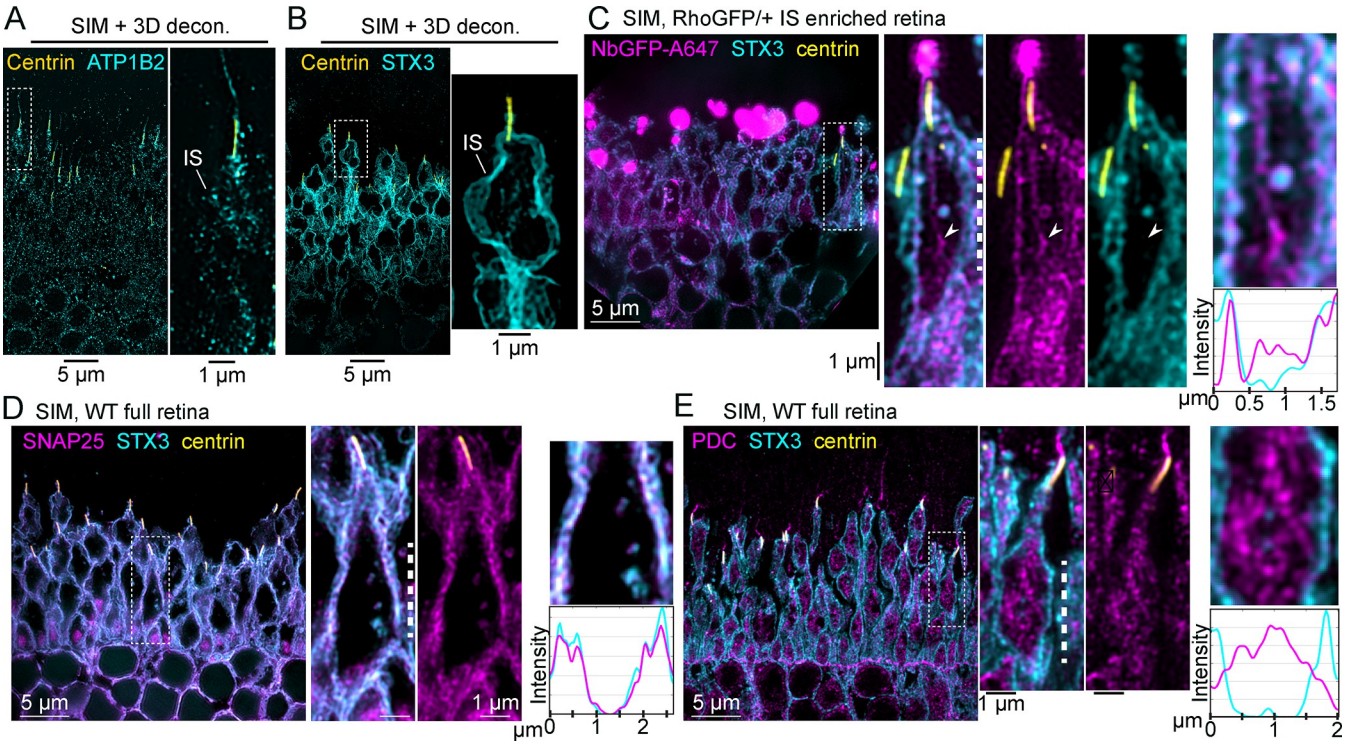

**Fig 4. Rho colocalized with STX3 at the rod IS plasma membrane.** (**A**) Example SIM z-projection image (processed with 3D deconvolution) of a thin plastic section from a WT mouse retina immunolabeled with antibodies for anti-centrin (yellow), to label the CC and BB, and anti-Na,K ATPase B2 subunit (ATP1B2, cyan) to label the IS plasma membrane. (**B**) Example SIM z-projection image with 3D deconvolution of a thin plastic section from a WT mouse retina immunolabeled with anti-centrin and anti-STX3 (cyan) to label the IS plasma membrane. (**C**) SIM image of an IS-enriched Rho-GFP/+ retina immunolabeled for NbGFP-A647 (magenta), STX3 (cyan), and centrin (yellow). In the magnified single rod example, an arrowhead indicates a cytoplasmic IS region with Rho-GFP fluorescence and no STX3 fluorescence. To demonstrate Rho-GFP colocalization with STX3 at the IS plasma membrane, a row average intensity plot is shown for a portion of the IS from the magnified single rod example marked with a white dashed line. SIM images of (**D**) SNAP25 (magenta) and (**E**) PDC (magenta) immunolabeling in WT full retina sections that are each colabeled with STX3 (cyan) and centrin (yellow) antibodies. For both magnified single rod examples, row intensity plots are provided for the IS regions marked by white dashed lines. Fluorescence intensities are normalized on all plots for clarity. Throughout the figure, magnified regions are indicated with a dashed box. Some magnified images are rotated so that the OS end of the rod is at the top of the image. Scale bar values match adjacent panels when not labeled. Numerical values corresponding to graphical data are provided in Table A in S1 Data. BB, basal body; CC, connecting cilium; IS, inner segment; NbGFP-A647, GFP nanobody Alexa 647 conjugate; OS, outer segment; PDC, phosducin; Rho, rhodopsin; SIM, structured illumination microscopy; STX3, syntaxin 3; WT, wild-type.

of the IS plasma membrane. In SIM images, RhoGFP + STX3 fluorescent signals were partially overlapped in the rod IS, including a near colocalization at the plasma membrane (Fig 4C), a result that was recapitulated for Rho-GFP-1D4/+ mice (S2C Fig), confirming Rho-GFP/Rho-GFP-1D4 localization at the IS plasma membrane. In both cases, internal Rho-GFP and Rho-GFP-1D4 did not completely colocalize with STX3. As a positive control for IS plasma membrane colocalization, immunolabeling of SNAP25, another IS SNARE complex protein like STX3 [54,55], was used in full WT retinas for SIM. In single rods from these retinas, STX3 and SNAP25 completely colocalized at the plasma membrane (Fig 4D). As a cytoplasmic protein control, immunolabeling of phosducin (PDC), a soluble IS chaperone protein [70], was used in WT retinas for SIM. In this case, PDC within single rod ISs was completely internal, corresponding to its cytoplasmic localization, and did not colocalize with STX3 at the plasma membrane (Fig 4E). Importantly, the correct and unperturbed localization of SNAP25, PDC, and STX3 in these data demonstrates the cellular preservation of the rod IS in retinas that are processed for immunofluorescence for either SIM or STORM.

Although NbGFP-A647 in Rho-GFP knock-in mice provides specific and rigorous Rho labeling, the endogenous mouse Rho localization pattern was also tested in the ISs of WT mouse rods. In IS-enriched WT retinas that were immunolabeled with Rho-C-1D4 antibody, the endogenous Rho localization pattern matched the Rho-GFP IS pattern, including localization at the IS plasma membrane and CC membrane, in addition to prominent bright puncta labeling in the myoid IS (Fig 5A). Mouse retinal immunolabeling for the *cis*-Golgi marker GM130 [71] was previously shown to label a bright, amorphous punctum in the IS myoid [53,72,73], a localization pattern that was recapitulated here with SIM (Fig 5B). IS-enriched Rho-GFP/+ retinas were co-immunolabeled with NbGFP-A647, GM130, STX3, and centrin antibodies to attempt to colocalize the bright myoid Rho puncta with GM130. However, in SIM images from these retinas, the Rho-GFP puncta did not directly colocalize with GM130. Instead, the Rho-GFP fluorescence closely surrounded the GM130+ *cis*-Golgi puncta in a reticulated fashion (Fig 5C).

Next, WT IS-enriched retinas were immunolabeled with either Rho-C-1D4 or Rho-N-4D2 along with STX3 to confirm endogenous mouse Rho localization at the IS plasma membrane with SIM. In each case, Rho colocalization with STX3 at the IS plasma membrane was observed along with a discontinuous internal labeling in the IS that was dense in the myoid IS (Figs 5D, 5E, S2D, and S2E). Overall, the localization of endogenous mouse Rho in the IS closely matched the Rho-GFP and Rho-GFP-1D4 IS localization pattern in Rho-GFP/+ and Rho-GFP-1D4 rods, respectively.

## STORM spatial analysis of Rho subcellular localization in mouse rod inner segments

STORM was used to generate single-molecule reconstructions of Rho localization in the IS from IS-enriched retinal samples. Molecule coordinates from STORM reconstructions of rod ISs were extracted to quantitatively test if a significant fraction of Rho molecules in the IS localized at the IS plasma membrane using a spatial localization analysis to measure the distance of Rho STORM molecules to the IS boundary or "hull." In all STORM data, reconstructed molecules from STX3 immunolabeling were used to manually define the rod IS hull, and centrin-2 was used as a marker of the CC and BB. Only rod ISs that contained a centrin-2-positive CC/BB were included in the analysis for consistency. The results, as detailed with statistical analysis below, strongly support the conclusion that Rho-containing internal membranes are preferentially localized near the IS plasma membrane.

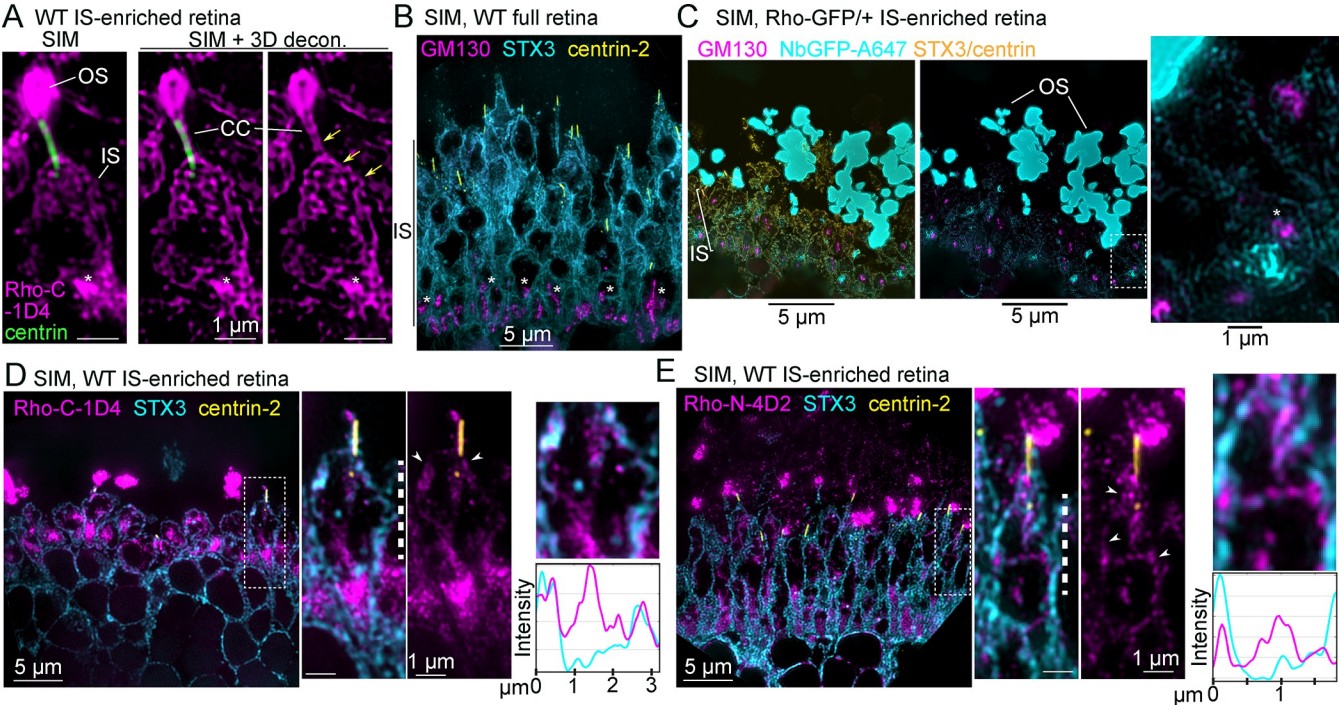

**Fig 5. Endogenous IS mouse Rho is located near the *cis*-Golgi and at the plasma membrane.** (**A**) SIM z-projection images of a single rod example from an IS-enriched, WT retina section immunolabeled with Rho-C-1D4 (magenta) and centrin antibody (green). Rho-C-1D4 is localized along the boundaries of the IS and CC (yellow arrows). The OS is indicated. (**B**) SIM z-projection image of a WT full retina section immunolabeled with GM130 (magenta), STX3 (cyan), and centrin-2 (yellow) antibodies. Asterisks indicate strong GM130 puncta staining in the proximal IS/myoid region corresponding to the *cis*-Golgi in each rod IS. (**C**) SIM z-projection image of a Rho-GFP IS-enriched retina section immunolabeled with NbGFP-A647 (cyan), along with GM130 (magenta), STX3 (yellow), and centrin (yellow) antibodies. The white arrow indicates a magnified portion of the IS layer, in which an asterisk indicates a GM130 punctum located directly near a Rho-GFP punctum in the myoid of a rod. (**D**, **E**) SIM images of WT IS-enriched retinas immunolabeled with either (**D**) Rho-C-1D4 antibody (magenta) or (**E**) Rho-N-4D2 antibody (magenta); both are colabeled with STX3 (cyan) and centrin-2 (yellow) antibodies. Partial endogenous Rho colocalization with STX3 at the IS plasma membrane is indicated with white arrowheads, and row average intensity plots are from the magnified portion of the IS corresponding to the region in single rod example images marked with a dashed line. Fluorescence intensities are normalized on both plots for clarity. Throughout the figure, magnified regions are indicated with either a dashed box or a dashed white arrow. Some magnified images are rotated so that the OS end of the rod is at the top of the image. Scale bar values match adjacent panels when not labeled. Numerical values corresponding to all graphical data are provided in Table B in S1 Data. CC, connecting cilium; IS, inner segment; NbGFP-A647, GFP nanobody Alexa 647 conjugate; OS, outer segment; Rho, rhodopsin; SIM, structured illumination microscopy; STX3, syntaxin 3; WT, wild-type.

First, STORM reconstructions from Rho-GFP/+ IS-enriched retinal sections immunolabeled with NbGFP-A647 were analyzed, and single rod ISs were identified. STORM molecules within the STX3+ IS hull were isolated from the STORM reconstruction of the retina section and plotted separately (Figs 6A and S3). In isolated STORM plots, Rho-GFP molecules partially overlapped with STX3 molecules in most examples, including at the IS hull. For comparison, a random distribution of molecules was plotted within the same IS hulls. Based on these plots, distance-to-hull values, which are the nearest distance of each molecule to the IS hull, were calculated for the Rho-GFP, STX3, and the random STORM molecules from each rod IS. Based on these data, a large fraction of Rho-GFP molecules were located within 0.2 μm of the IS hull, with distributions typically matching STX3 molecules as opposed to random molecule distributions (Figs 6A and S3). In alternate rod examples where Rho molecules are more densely localized internally, the distribution of Rho-GFP distance-to-hull values shifted to higher mean distances (S3A Fig); however, Rho-GFP reconstructed molecules from these STORM plots are still consistently localized along the IS hull. The STORM analysis was repeated in Rho-GFP-1D4/+ retinas using NbGFP-A647 labeling, and the distributions of

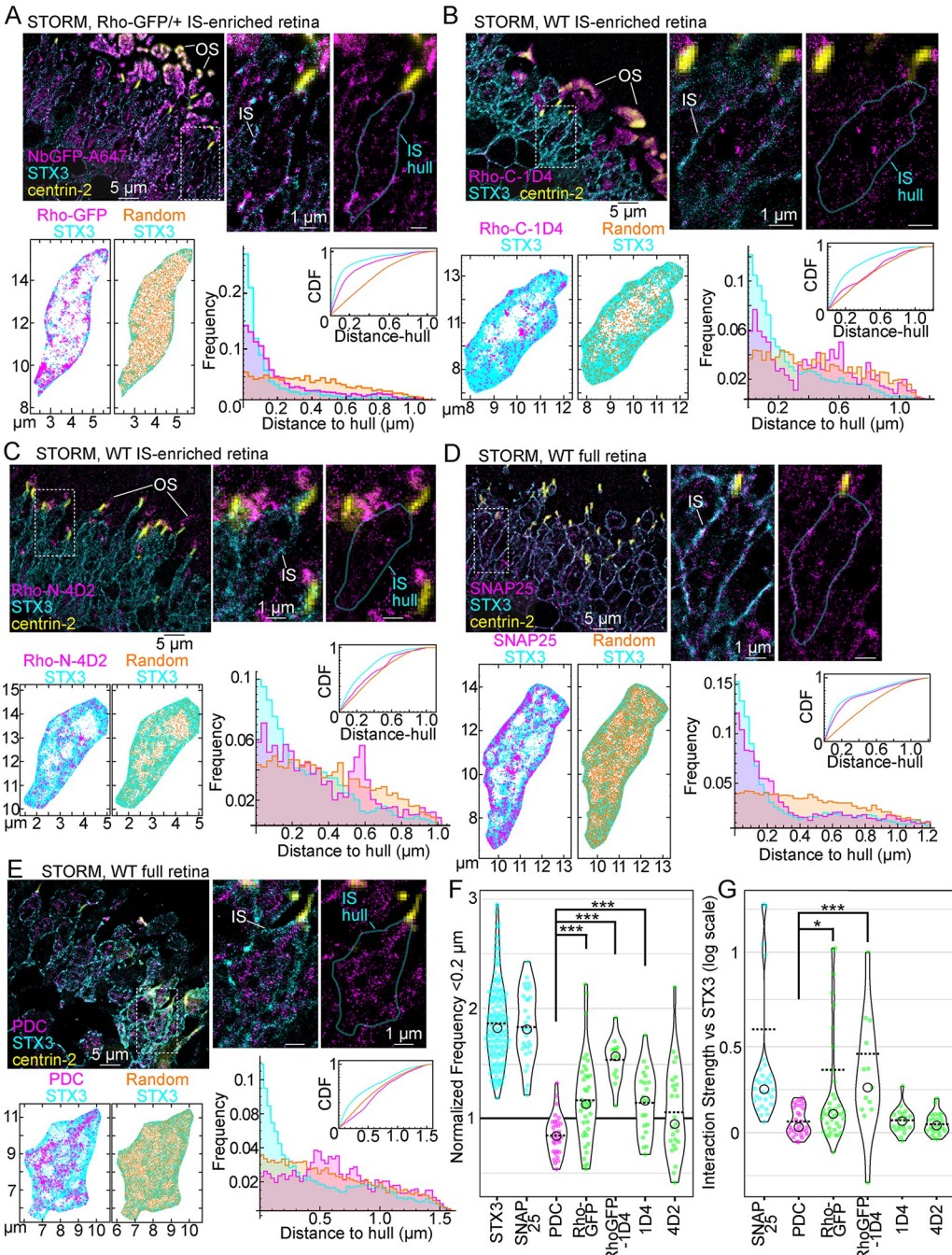

**Fig 6. STORM spatial analysis of Rho localization in rod ISs.** (**A**) STORM reconstruction of an IS-enriched Rho-GFP/
+ retina immunolabeled with NbGFP-A647 (magenta) and STX3 (cyan) and centrin-2 (yellow) antibodies. OSs are
indicated. The centrin-2+ widefield images are superimposed on STORM reconstruction images to mark the positions of
rod CCs and BBs. In single rod examples, the IS region is indicated, and the IS hull—the manually defined STX3+ IS
boundary—is outlined in cyan in a duplicate image with the STX3 STORM reconstruction removed. From the rod
example in (**A**), the Rho-GFP and STX3 STORM molecule coordinates within the IS hull are plotted (Rho-GFP
molecules = 4,239, STX3 molecules = 5,215). In the adjacent plot, a random distribution of coordinates within the IS hull
matching the number of Rho-GFP molecules (4,239) are plotted in orange with the STX3 molecules. Nearest distance-to-
hull measurements for Rho-GFP, STX3, and random molecules are plotted in a frequency graph and a CDF graph.
Colors in the graphs match the molecule plots. (**B**-**E**) STORM reconstructions of IS-enriched WT retinas immunolabeled
with either Rho-C-1D4 antibody (**B**) or Rho-N-4D2 (**C**), and full WT retinas immunolabeled with (**D**) SNAP25 or (**E**)
PDC; all are co-immunolabeled with STX3 and centrin-2 antibodies. Single rod examples and IS hulls are indicated.
Molecules coordinates are plotted along with corresponding randomly plotted molecules (orange) as in (**A**), and

distance-to-hull measurements are plotted as frequency and CDF graphs. Molecule counts: (**B**) Rho-C-1D4 = 1,430, STX3 = 24,596, Random = 1,430; (**C**) Rho-N-4D2 = 3,199, STX3 = 24,044, Random = 3,199; (**D**) SNAP25 = 7,871, STX3 = 28,817, Random = 7,871; (**E**) PDC = 14,556, STX3 = 34,544, Random = 14,556. Dashed boxes indicate the single rod examples in magnified images. Scale bar values match adjacent panels when not labeled. (**F**) For each rod analyzed with STORM, the distance-to-hull frequency within 0.2 μm was divided by the distance-to-hull frequency within 0.2 μm of the corresponding random coordinates to acquire "normalized frequency <0.2 μm" values, which are compared as violin plots. STX3 *n* value (for number of rods analyzed) = 162, SNAP25 *n* = 31, PDC *n* = 31, Rho-GFP *n* = 40, Rho-GFP-1D4 *n* = 13, Rho-C-1D4 *n* = 21, and Rho-N-4D2 *n* = 25 conditions were tested for statistical significance using the Mann–Whitney *U* test. PDC vs. Rho-GFP ***P value < 0.001; PDC vs. Rho-GFP-1D4 ***P value < 0.001; PDC vs. 1D4 ***P < 0.001. (**G**) The same STORM data were used to perform Mosaic interaction analyses to test the colocalization between STX3 molecules and the other immunolabeled targets from the same rod ISs. Interaction strength values are compared as violin plots on a log scale. PDC vs. Rho-GFP, Rho-C-1D4, and Rho-N-4D2 (same *n* values as (**F**)) were tested for statistical significance using the Mann–Whitney *U* test. PDC vs. Rho-GFP *P = 0.026; PDC vs. Rho-GFP-1D4 ***P < 0.001. In violin plot graphs, circles = median values and dashed lines = mean values. Numerical values corresponding to all graphical data are provided in Table C in S1 Data. BB, basal body; CC, connecting cilium; CDF, cumulative distribution function; IS, inner segment; NbGFP-A647, GFP nanobody Alexa 647 conjugate; OS, outer segment; PDC, phosducin; Rho, rhodopsin; STORM, stochastic optical reconstruction microscopy; STX3, syntaxin 3; WT, wild-type.

Rho-GFP-1D4 distance-to-hull values were also consistently weighted toward lower mean distances (S3B Fig).

Next, the same STORM spatial analysis was performed in WT IS-enriched retinas immunolabeled with Rho-C-1D4 and Rho-N-4D2 antibodies (Figs 6B, 6C, S3C, and S3D). In these rod reconstructions, the distribution of Rho molecule distance-to-hull measurements varied based on the internal distribution of Rho in any given rod, but as with the Rho-GFP data, endogenous Rho molecules were always localized to some degree near the IS hull, typically as a large fraction of molecules localized within 0.2 μm of the IS hull.

STORM IS molecule coordinates from SNAP25 and PDC immunolabeled WT full retina samples were also analyzed as spatial analysis controls. As with our SIM data, SNAP25 molecules colocalized with STX3 molecules in rod STORM reconstructions, and SNAP25 distance-to-hull distributions also closely matched STX3 (Figs 6D and S3E). PDC IS reconstructed STORM molecules, on the other hand, were almost completely internal, and distance-to-hull measurements accumulated at greater distances from the IS hull, completely distinct from STX3 distance distributions from the same rod examples (Figs 6E and S3F).

All distance-to-hull measurements were aggregated from this STORM rod IS spatial analysis, including for STX3. The frequency of molecules <0.2 μm from the IS hull for any STORM target (e.g., Rho, STX3) was normalized to the corresponding random <0.2 μm frequency from the same rod reconstruction to account for different IS areas and labeling densities in our STORM data. Normalized frequency values were plotted for comparison (Fig 6F), and STX3 and SNAP25 normalized values were clearly greater than 1 in aggregate, confirming a greater than random localization of STX3 and SNAP25 within 0.2 μm of the IS hull. Rho-GFP, Rho-GFP-1D4, Rho-C-1D4, and Rho-N-4D2 normalized values all were, in aggregate, nearer to 1 due to the variable fraction of internal Rho IS molecules that were also measured in this analysis. However, one-sample *t* tests were performed for all Rho conditions to statistically compare the normalized data to 1. Based on this test, Rho-GFP, Rho-GFP-1D4, and Rho-C-1D4 data were statistically different compared to 1 (two-tailed *P* values: Rho-GFP versus 1 *P* = 0.0117, Rho-GFP-1D4 versus 1 *P* < 0.0001, Rho-C-1D4 versus 1 *P* = 0.0349), confirming that the fraction of Rho molecules in those rods localized within 0.2 μm of the IS hull is larger than expected for randomly distributed molecules. Furthermore, by direct comparison, Rho-GFP, Rho-GFP-1D4, and Rho-C-1D4 normalized <0.2 μm frequency values were significantly greater than cytoplasmic PDC <0.2 μm frequency values (Fig 6F). Normalized frequency values of distance-to-hull measurements within 0.1 μm and 0.3 μm were also aggregated for all

conditions, and the same general distribution of the aggregated data was observed (S4A and S4B Fig). In summary, despite a variable fraction of internally localized Rho in STORM-reconstructed rods, most rod ISs contain a nonrandom fraction of Rho molecules that localized near the IS plasma membrane within 0.1 to 0.3 μm from the IS hull with a greater frequency compared with internally localized STORM PDC molecules.

The same rod IS STORM data were used for a Mosaic spatial interaction analysis of colocalization [74] comparing the spatial distribution patterns of STX3 versus SNAP25, PDC, Rho-GFP, Rho-GFP-1D4, Rho-C-1D4, and Rho-N-4D2 within our manually defined IS hull regions. Interaction strength values for these comparisons were calculated and normalized to the interaction strength values of random molecule distributions (each compared to STX3) to account for rod-to-rod variations in IS area and labeling density. When aggregated, SNAP25 versus STX3 had the greatest normalized interaction strength scores, as expected, while both Rho-GFP versus STX3 and Rho-GFP-1D4 versus STX3 normalized interaction strength values were significantly greater than PDC versus STX3 (Fig 6G), indicating a strong colocalization between Rho-GFP/Rho-GFP-1D4 and STX3 in the STORM data. Rho-C-1D4 and Rho-N-4D2 interaction strength values compared to STX3 were not statistically different from PDC likely due to larger internal Rho densities in these rods that do not colocalize with STX3.

The finding that Rho localized to the rod IS hull/plasma membrane was tested using a surface labeling (no detergent) variation of our whole-retina immunolabeling protocol for SIM and STORM. Wheat germ agglutinin (WGA), which labels the rod glycocalyx [75], was used as a general cell surface marker. In retinal sections from WT, Rho-GFP/+, and Rho-GFP-1D4/+ mice, WGA labeling was prominent in the OS layer and at the outer limiting membrane at the boundary between the IS and ONL (Fig 7A–7C). Thus, WGA enabled the identification of the IS layer in sections of the outer retina. Next, Rho-N-4D2 immunolabeling was used in surface labeled retinas to target the extracellular N-terminus of Rho, which would be exposed to surface labeling for Rho proteins inserted at the IS plasma membrane. With SIM, we found positive fluorescent Rho-N-4D2 surface labeling throughout the IS layer (Fig 7D–7F), in addition to the expected dense Rho N-terminus surface labeling of the OS plasma membrane [76]. Rho surface labeling specificity was confirmed by the lack of signal from Rho-C-1D4 surface labeling of WT retinas (Fig 7G). STORM was used to statistically compare the surface localization of Rho-N-4D2 in rods from WT, Rho-GFP/+, and Rho-GFP-1D4/+ mice. In single rods, STORM molecular coordinates for Rho-N-4D2 outlined ISs in a surface labeling pattern (Fig 7H–7J) and were extracted for a distance-to-hull spatial analysis. The frequencies of IS surface-localized molecules within 0.2 μm of the IS hull, as normalized to a random distribution of molecules for each rod IS, were statistically similar between genotypes (Fig 7K).

Immuno-EM was also performed to validate our STORM single-molecule localization findings, using another variation of the same whole retina immunolabeling protocol. Using this method, Nanogold-labeled STX3 was accumulated at the rod IS plasma membrane (S4C Fig), while Rho-C-1D4 Nanogold labeling in rods from WT IS-enriched retina also aligned to the IS plasma membrane border (S4D Fig). Ultrastructural damage in immuno-EM retina samples could not be avoided due to the harsh immuno-EM processing steps that have been previously described [77–79]; however, these EM-based localization results support our super-resolution fluorescence localization of Rho IS at the IS plasma membrane in mouse rods.

Coimmunoprecipitation (co-IP) experiments were performed using IS-enriched retina and Triton X-100 detergent extraction to identify IS Rho protein–protein interactions in vivo to support our STORM colocalization data. First, using Rho-GFP/+ IS-enriched retinal lysates, co-IPs were performed with GFP-Trap magnetic beads to capture Rho-GFP protein, which immunoprecipitated with both STX3 and SNAP25 (Fig 8A). GC1 (guanylate cyclase 1), a known protein interactor with Rho [80], was the positive control, and PDC was the negative

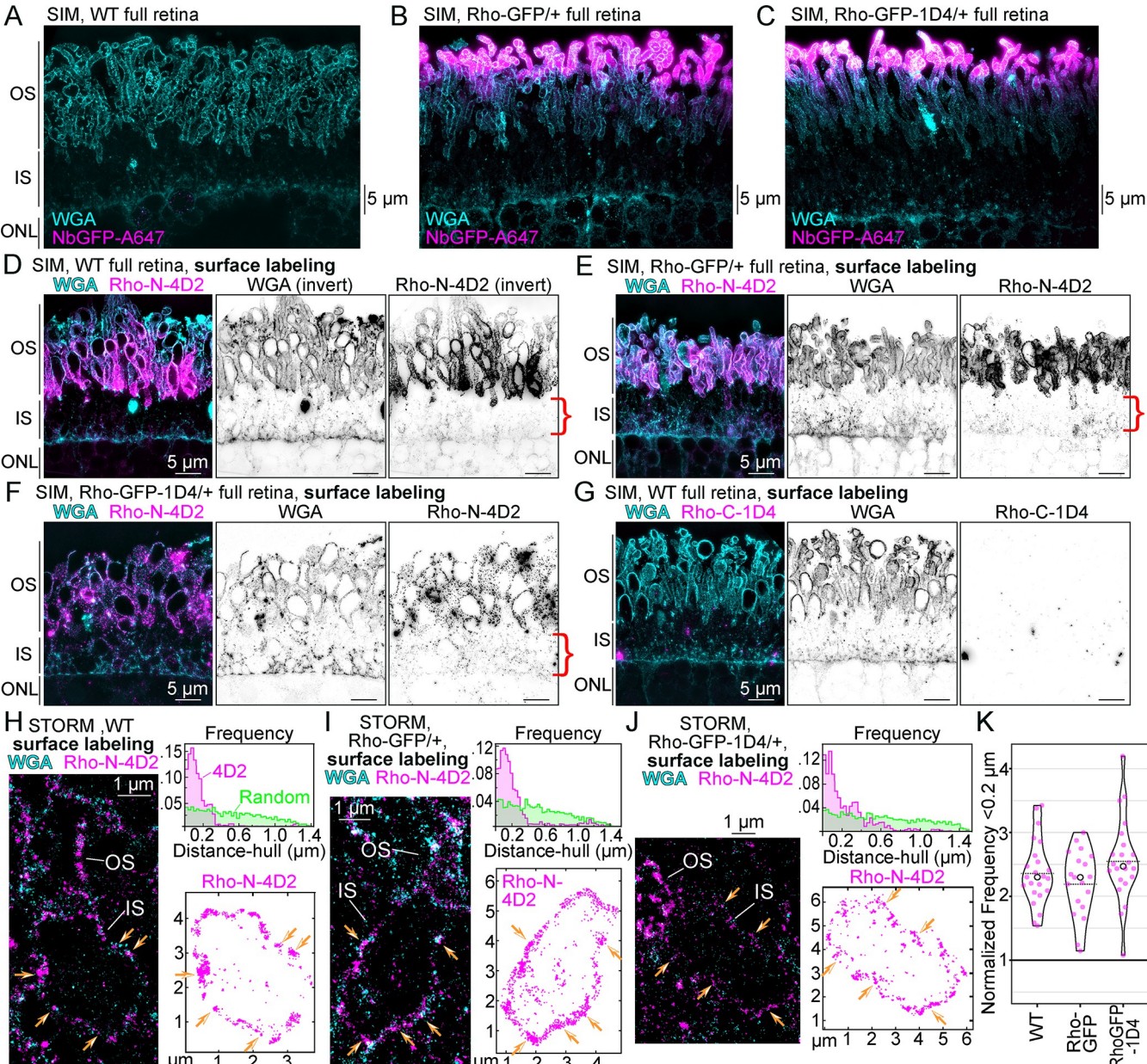

**Fig 7. Super-resolution mapping of Rho IS surface labeling in mouse rods.** Z-projection SIM images of thin plastic sections of full (unpeeled) retinas from (**A**) WT, (**B**) Rho-GFP/+, and (**C**) RhoGFP-1D4 mice that were all labeled with WGA (cyan) and NbGFP-A647 (magenta) using normal, permeabilizing immunostaining conditions. Note that NbGFP-A647 only labels Rho-GFP/+ and Rho-GFP-1D4/+ rod OS tips as in Fig 1E and 1F. (**D–F**) Z-projection SIM images from full retinas labeled with WGA (cyan) and Rho-N-4D2 antibody (magenta) from WT (**D**), Rho-GFP/+ (**E**), or Rho-GFP-1D4/+ (**F**) mice using a surface labeling (no fixative, no detergent) protocol (see Materials and methods). Inverted WGA and Rho-N-4D2 single channel images are shown adjacent to the composite image. Red brackets indicate positive Rho-N-4D2 fluorescent signal in the IS. (**G**) Z-projection SIM image from a full WT retina labeled with WGA (cyan) and Rho-C-1D4 antibody (magenta). Lack of Rho-C-1D4 surface immunolabeling demonstrates Rho-N-4D2 extracellular labeling specificity and effectiveness. The OS, IS, and ONL layers, based on the WGA labeling pattern, are indicated for each SIM image. Scale bar values match adjacent panels when not labeled. (**H–J**) STORM reconstruction examples of rods from WT (**H**), Rho-GFP/+ (**I**), or Rho-GFP-1D4/+ (**J**) full retinas that were surface labeled with WGA (cyan) and Rho-N-4D2 antibody (magenta). The OS and IS regions are indicated. Adjacent to each STORM image is the Rho-N-4D2 molecular coordinates for the IS region. The number of Rho-N-4D2 molecules in (**H**) = 2,257, in (**I**) = 2,022, and in (**J**) = 2,764. Surface localization of Rho-N-4D2 molecules is indicated with orange arrows. Nearest distance-to-hull measurements for Rho-N-4D2 (magenta) and the corresponding set of random molecules (green) are plotted in a frequency graph above the molecule plots. For each rod analyzed with STORM, the distance to hull frequency within 0.2 μm was divided by the distance to hull frequency within 0.2 μm of the corresponding random coordinates to acquire "normalized frequency <0.2 μm" values, which are compared as violin plots grouped by genotype (**K**). WT *n* value (for number of rods analyze) = 23, "Rho-GFP" for Rho-GFP/+ rods *n* = 19, "Rho-GFP-1D4" for Rho-GFP-1D4/+ rods *n* = 22. No significant difference in these data were based on a one-way ANOVA test (*P* value = 0.144). In the violin plot graph, circles = median values and dashed lines = mean values. Numerical values corresponding to all graphical data are provided in Table D in S1 Data. IS,

inner segment; NbGFP-A647, GFP nanobody Alexa 647 conjugate; ONL, outer nuclear layer; OS, outer segment; Rho, rhodopsin; SIM, structured illumination microscopy; STORM, stochastic optical reconstruction microscopy; WGA, wheat germ agglutinin; WT, wild-type.

control. Similar co-IP experiments using IS-enriched Rho-GFP-1D4 retinal lysates and GFP-Trap beads yielded the same interaction results as the Rho-GFP pulldown (Fig 8B). Next, anti-1D4 IgG-bound agarose beads were used to capture endogenous mouse Rho from IS-enriched Rho-GFP/+ lysates using the same co-IP controls as Fig 8A and 8B. Here, Rho-GFP protein was immunoprecipitated, demonstrating that the endogenous mouse Rho interacts with human Rho-GFP in Rho-GFP/+ rods (Fig 8C). STX3 was also immunoprecipitated with endogenous mouse Rho. Finally, using WT IS-enriched retinas and anti-1D4 IgG beads, STX3 was immunoprecipitated with Rho (Fig 8D), demonstrating this protein–protein interaction in vivo.

## Rhodopsin surrounds the distal appendages at the inner segment–connecting cilium junction

Since we observed a continuous string of Rho fluorescence between the IS plasma membrane and into the CC membrane in mouse rods (Figs 3B and 5A), we next tested Rho molecule localization relative to the DAPs, which are the BB structures at the IS/CC interface in rods. The DAPs are 9 proteinaceous blades that project radially from the mother centriole at the base of the CC and extend to the plasma membrane [30]. In mammalian primary cilia, the DAPs were shown to function as binding sites for IFT (intraflagellar transport) proteins,

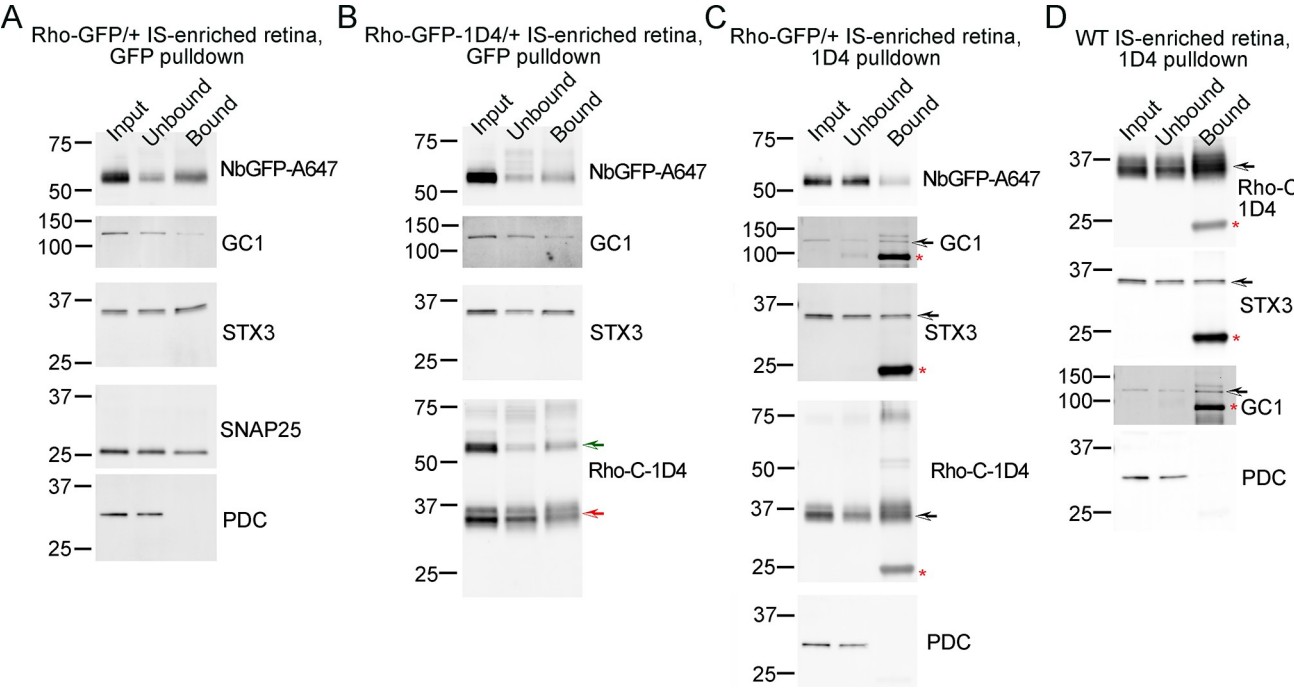

**Fig 8. STX3 coimmunoprecipitates with Rho in vivo.** Co-IP results from (**A**) Rho-GFP/+ IS-enriched retinal lysates incubated with GFP-Trap agarose beads, (**B**) Rho-GFP-1D4/+ IS-enriched retinal lysates incubated with GFP-Trap agarose beads, (**C**) Rho-GFP/+ IS-enriched retinal lysates incubated with anti-1D4 IgG agarose beads, and (**D**) WT IS-enriched retinal lysates incubated with anti-1D4 IgG agarose beads. In all western blots, input lanes correspond to 2% (% vol/vol) of the starting lysate volume, unbound lanes correspond to 2% (% vol/vol) of lysate volume post bead incubation, and bound lanes correspond to half the total eluate from each co-IP. Antibodies/nanobodies used for immunodetection are listed to the right of each corresponding western blot scan. Molecular weight marker sizes are indicated in kDa. Black arrows mark the correct size bands when other bands are present on the scan. Red asterisks indicate mouse IgG bands. In (**B**), the red arrow indicates the endogenous Rho band and the green arrow indicates the Rho-GFP-1D4 band. co-IP, coimmunoprecipitation; IgG, immunoglobulin G; Rho, rhodopsin; STX3, syntaxin 3; WT, wild-type.

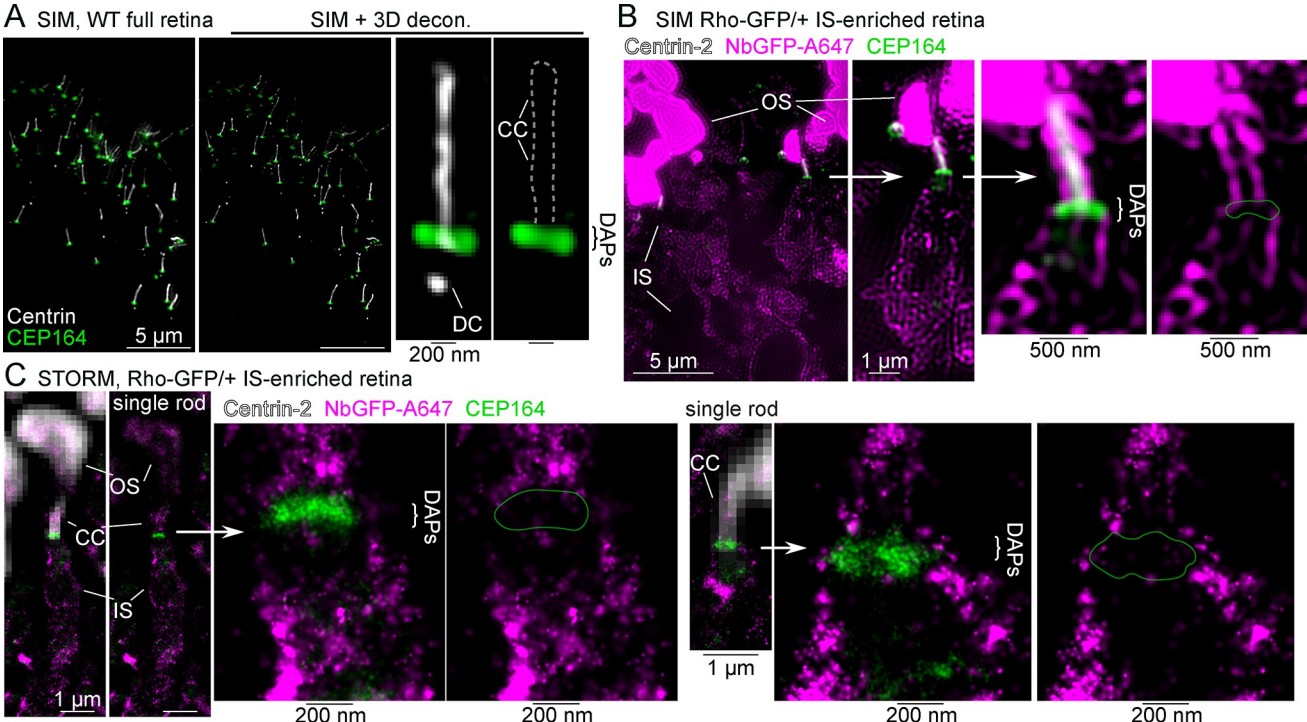

**Fig 9. Rho localizes around the DAPs in mouse rods.** (**A**) SIM z-projection images of a WT full retina section immunolabeling with a centrin antibody that marks the location of rod CC and the DC of the BB in white, along with a CEP164 antibody that labels the DAPs in green. The SIM retina image is also shown after 3D-deconvolution processing (SIM + 3D decon.), and a single rod cilium example is magnified. (**B**) SIM image from peeled Rho-GFP/+ mouse labeled with NbGFP-A647 (magenta), anti-CEP164 antibody (green), and anti-centrin antibody (white). A single rod is magnified, and the DAPs region is further magnified. The remaining OSs and the IS region are indicated. (**C**) STORM reconstruction single rod examples and magnified DAPs regions from Rho-GFP/+ IS-enriched retina immunolabeled with NbGFP-A647 (magenta) and a CEP164 antibody (green). Centrin-2 antibody labeling marks the position of the CC (white), and residual NbGFP-A647 signal labels the OS; both are captured as widefield fluorescence images. White arrows indicate the regions that are magnified in the adjacent image. Scale bar values match adjacent panels when not labeled. BB, basal body; CC, connecting cilium; DAP, distal appendage; DC, daughter centriole; NbGFP-A647, GFP nanobody Alexa 647 conjugate; IS, inner segment; OS, outer segment; Rho, rhodopsin; SIM, structured illumination microscopy; STORM, stochastic optical reconstruction microscopy; WT, wild-type.

which facilitate active ciliary transport, and to function in the gating of the ciliary GPCRs Smoothened and SSTR3 [81].

Here, an antibody targeting the DAP protein CEP164 was used to mark the position of the DAPs in rods, and in SIM images, the CEP164+ DAPs were localized as a well-defined line of fluorescence at the proximal end of the CC, which is the IS-CC interface (Fig 9A). Next, Rho-GFP/+ IS-enriched retinas were immunolabeled with NbGFP-A647 along with CEP164 and centrin antibodies to localize Rho at the DAPs using SIM and STORM. In these data, Rho-GFP molecules were again localized as continuous strings of fluorescence between the IS and CC plasma membranes but in a pattern surrounding the CEP164+ DAP blades (Fig 9B and 9C) with only a few STORM molecules colocalized with the CEP164+ DAPs. These results suggest that Rho is not integrated into the DAPs but is likely associated with the surrounding plasma membrane between the IS and CC.

## Golgi and post-Golgi trafficking proteins are localized at the inner segment plasma membrane in mouse rods

We next tested the super-resolution localization of proteins that have previously been associated with the Rho secretory pathway in the rod IS. Rab11a, a post-Golgi small GTPase, was

shown to mediate Rho trafficking in post-Golgi vesicles in frog rods [47,48] and was previously localized to the IS in mouse retinas in a semi-diffuse, puncta-like pattern [52,73,82]. Here, WT retina cryosections were immunolabeled with a specific Rab11a antibody [83], and Rab11a was localized with confocal imaging to the IS, in the ONL surrounding the nuclei, and in the OPL (Fig 10A). With SIM, Rab11a was localized as bright puncta throughout the OS, IS, and ONL, and in individual rod ISs, many Rab11a+ puncta were observed to be colocalized with the STX3+ IS plasma membrane (Fig 10B). After puncta counting, the rate of IS plasma membrane associated Rab11a+ puncta was 45.8% ± 14.3% (SD) per mouse rod IS ($n$ = 955 puncta, $n$ = 32 rods); the rest of the Rab11a+ puncta were internal/cytoplasmic. In STORM reconstructions, Rab11a molecules were localized into molecule clusters within the IS, which were isolated for visualization using a Voronoi tessellation clustering algorithm (Fig 10C).

Next, although the GM130+ $cis$-Golgi was prominently localized in the myoid region of the IS (Fig 5B); the localization of Golgi alpha-mannosidase II (GMII), a $medial/trans$-Golgi glycoside hydrolase that was previously shown to process Rho protein in Golgi [84], was localized throughout the IS of mouse retinas using confocal microscopy (Fig 10D). In SIM images, GMII was also localized in discrete puncta within rod ISs, albeit less densely than Rab11a+ IS puncta (Fig 10E); however, more GMII+ puncta were colocalized with the IS plasma membrane. The rate of IS membrane associated GMII+ puncta was 73.8% ± 19.2% (SD) ($n$ = 303 puncta, $n$ = 27 rods). Using SIM with 3D deconvolution, the diameters (⌀) of single, isolated IS Rab11a+ and GMII+ puncta were measured. Based on diameter, Rab11a+ puncta were distributed in 2 groups: <200 nm ⌀ puncta (mean ⌀ = 126.7 nm ± 23.5 nm, $n$ = 34 puncta) and >200 nm ⌀ puncta (mean ⌀ = 320.8 nm ± 33.8 nm, $n$ = 34 puncta), while GMII+ puncta were normally distributed as 1 group (mean ⌀ = 119.6 nm ± 14.5 nm, $n$ = 58 puncta) (Fig 10F).

Together, these results indicate that vesicular Golgi and post-Golgi trafficking organelles are targeted to the IS plasma membrane in mouse rods. For comparison, cytoplasmic dynein-1 and the ciliary rootlet were also localized in the mouse IS with super-resolution fluorescence. Cytoplasmic dynein-1 is essential in rods as the putative motor complex for intracellular cytoplasmic transport. An antibody targeting the force-generating heavy chain of the dynein-1 complex, DYNC1H1, was used to localize dynein-1 throughout the entire rod IS layer, and with STORM, DYNC1H1 molecules were localized in a homogenous distribution throughout the IS (Fig 10G). An antibody targeting rootletin, the core protein of the ciliary rootlet, was used to reconstruct the rootlet in rods with STORM (Fig 10H).

## Rhodopsin colocalizes with Rab11a in rod inner segments

STORM was used to test the colocalization of Rho-GFP with Rab11a, DYNC1H1, and rootletin in IS-enriched Rho-GFP/+ retinas. In reconstructed mouse ISs, a fraction of Rho-GFP molecules were localized surrounding Rab11a+ molecule clusters (Figs 10I and S5A). Similarly, in IS-enriched Rho-GFP-1D4/+ rods, STORM-reconstructed Rho-GFP-1D4 molecules partially accumulated around Rab11a+ clusters in the IS (Figs 10J and S5B). Rho-GFP also partially colocalized with the more homogenously distributed DYNC1H1+ STORM molecules in the IS (Figs 10K and S5C); however, Rho-GFP was not consistently colocalized with the ciliary rootlet (Figs 10L and S5D). STORM colocalization from these mouse rod IS reconstructions was then quantified using the Mosaic spatial interaction analysis, and the interaction strength scores, normalized to random molecule distributions, were calculated (Fig 10M). From these aggregated scores, one-sample $t$ tests were again performed to compare the normalized data to 1. Using this test, Rab11a + Rho-GFP and Rab11a + Rho-GFP-1D4 data were statistically greater than 1 (two-tailed $P$ values: Rab11a + Rho-GFP versus 1 $P$ = 0.0089, Rab11a + Rho-GFP-1D4 versus 1 $P$ = 0.0051) indicating a nonrandom colocalization between Rho and Rab11a in the

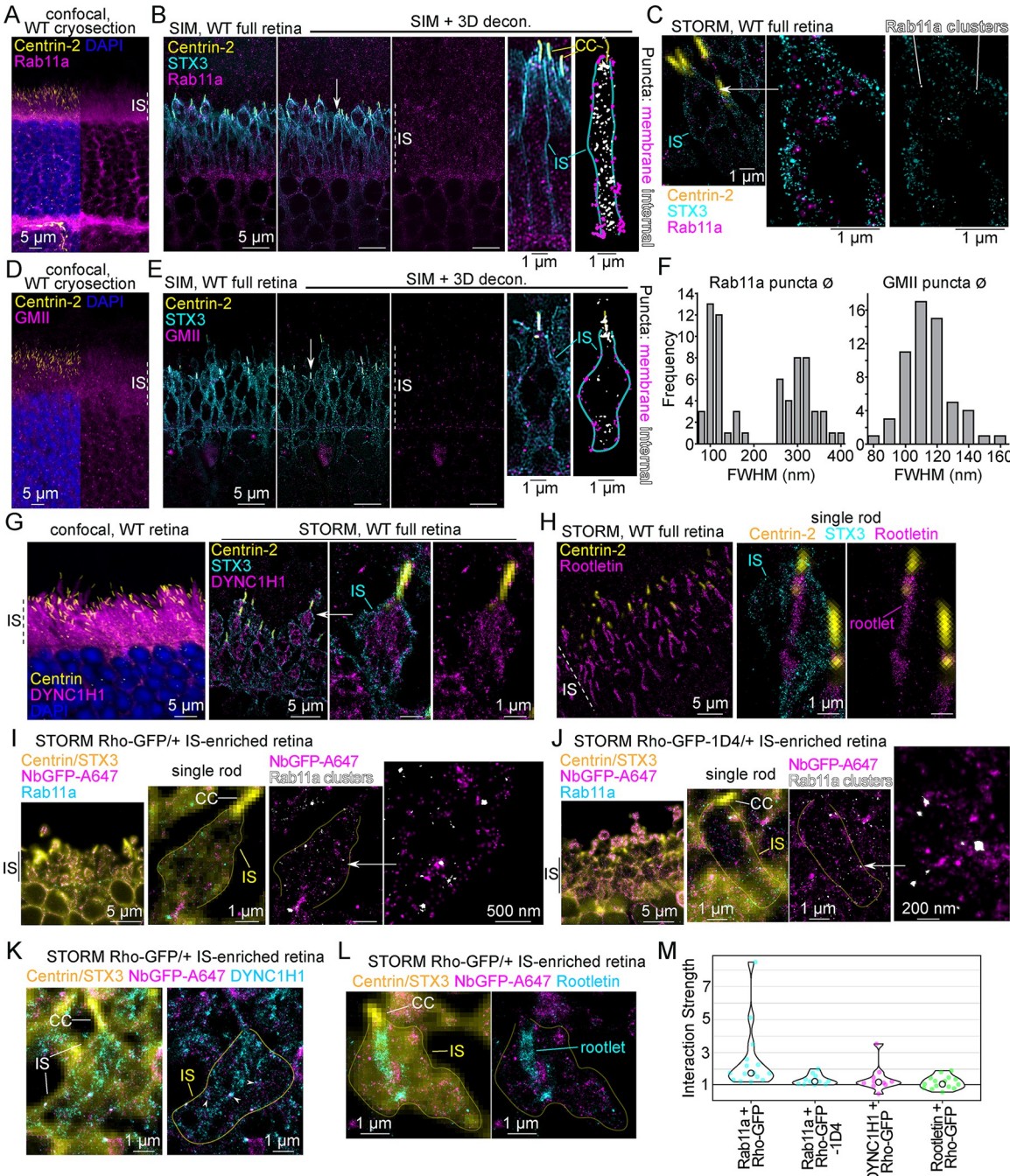

**Fig 10. Rho colocalizes with Rab11a in mouse rod ISs.** (**A**) Z-projection of a WT retinal cryosection co-immunolabeled with centrin-2 and Rab11a antibodies and counterstained with DAPI. (**B**) SIM z-projection image of a WT retina co-immunolabeled with centrin-2, STX3, and Rab11a antibodies. The SIM retina image is also shown after 3D-deconvolution processing (SIM + 3D decon.), and Rab11a + puncta are localized in the IS layer. A single rod example is magnified, and a threshold image of the Rab11a channel is shown pseudocolored to depict puncta that are localized at the STX3+ IS membrane hull as magenta, and puncta that are localized internally or at the CC as white. (**C**) STORM reconstruction of a single rod from a WT retina immunolabeled as in (**B**). A magnified region is shown, and in the adjacent image, Rab11a+ clusters identified with Voronoi tessellation (see Materials and methods) are in white. (**D**) Z-projection of a WT retinal cryosection co-immunolabeled with centrin-2 and GMII antibodies and counterstained with DAPI. (**E**) SIM images, including 3D decon. processed images, of a WT retina co-immunolabeled with centrin-2, STX3, and GMII antibodies. As in (**B**), a single rod example is shown, and in the adjacent image, membrane-localized GMII+ puncta are pseudocolored magenta, and internal (and ciliary) puncta are white. (**F**) Frequency plots for FWHM measurements of individual Rab11a and GMII IS-localized puncta from the SIM data represented in (**B**) and (**E**). For Rab11a FWHM values, *n* (number of puncta) = 67. For GMII FWHM values, *n* (number of puncta) = 58. (**G**) Confocal z-projection image of a WT retina cryosection immunolabeled with centrin and DYNC1H1 antibodies, as

well as DAPI counterstaining. DYNC1H1+ fluorescence fills the IS layer. In the adjacent panel, a STORM reconstruction of a WT retina immunolabeled with centrin-2, STX3, and DYNC1H1 antibodies is depicted. A single rod example is shown. (**H**) STORM reconstruction of a WT retina immunolabeled with centrin-2, STX3, and rootletin antibodies. A single rod example is shown, and the ciliary rootlet is indicated. (**I-M**) STORM images of a (**I**) Rho-GFP/+ IS-enriched retina or (**J**) Rho-GFP-1D4/+ IS-enriched retina, each co-immunolabeled with NbGFP-A647, and centrin, STX3, and Rab11a antibodies. STORM reconstruction channels—NbGFP-A647 (magenta) and Rab11a (cyan)—are superimposed with the matching widefield fluorescence image of centrin/STX3 immunolabeling (combined, yellow). IS regions are indicated. A single rod example is shown, and the CC is indicated. In the adjacent image, Rab11a + clusters identified with Voronoi tessellation are in white, and the STX3+ IS hull is outlined in yellow. A white arrow indicates a further magnified region of Rho-GFP molecules localized around Rab11a clusters; however, there was a relatively low degree of colocalization between Rho and Rab11a. Next, Rho-GFP was co-immunolabeled with (**K**) DYNC1H1 antibody or (**L**) Rootletin antibody. In both, the locations of the CC and the IS outline are indicated, and in (**K**) areas where Rho-GFP and DYNC1H1 molecules overlap are indicated with white arrowheads. In (**L**), the location of the ciliary rootlet is indicated. (**M**) STORM data from (**I-L**) conditions were used to perform Mosaic interaction analyses to test the colocalization between Rho-GFP molecules and the other immunolabeled target from the same rod IS. Interaction strength values are compared as violin plots (circles = median values and dashed lines = mean values). *N* values, corresponding to the number of rods from each condition, are Rab11a vs. Rho-GFP, *n* = 15; Rab11a vs. Rho-GFP-1D4, *n* = 11; DYNC1H1 vs. Rho-GFP, *n* = 10; Rootletin vs. Rho-GFP, *n* = 15. In all panels, white arrows indicate regions that are magnified. Scale bar values match adjacent panels when not labeled. Numerical values corresponding to all graphical data are provided in Table E in S1 Data. CC, connecting cilium; FWHM, full width half maximum; GMII, Golgi alpha-mannosidase II; IS, inner segment; NbGFP-A647, GFP nanobody Alexa 647 conjugate; Rho, rhodopsin; SIM, structured illumination microscopy; STORM, stochastic optical reconstruction microscopy; STX3, syntaxin 3; WT, wild-type.

IS. Although we observed partial colocalization between Rho-GFP and DYNC1H1 with STORM, the overall abundant and diffuse pattern of IS DYNC1H1 is likely why the interaction strength with Rho molecules was not distinguishably higher than random.

## Discussion

In this study, we developed an optimized approach for localizing Rho in the IS of mouse rod photoreceptor cells. Rho is the most abundant protein in the membrane discs of rod OS cilia [3], which are constantly renewed through a robust network of protein translation and protein trafficking pathways in the IS. Based on our localization findings, including single-molecule localization and spatial analysis data, we established that Rho localized prominently at the IS plasma membrane, at the surface, in an even distribution along the entire boundary of the IS, including along the ellipsoid and myoid. IS plasma membrane Rho formed a string of localized fluorescence that was continuous with the CC plasma membrane. Rho colocalized at the IS plasma membrane with the t-SNARE protein STX3, and a significant fraction of Rab11a-positive and GMII-positive fluorescent puncta, as putative Golgi and post-Golgi vesicles, also localized at the IS hull. Combined with our partial colocalization of Rho with Rab11a, these findings suggest that after Golgi processing, new Rho proteins are trafficked within vesicles to the IS plasma membrane as a target/destination membrane. S5E Fig is a diagram of this model that also depicts Rho trafficking to the CC and then ultimately to the OS base within the IS plasma membrane through a lateral transport mechanism.

Such a lateral transport mechanism has been previously described for ciliary GPCRs in cultured mouse cells, in which Smoothened and the D1-type dopamine receptor proteins were shown to transport laterally between the plasma membrane and the primary cilia [85,86]. Furthermore, dense rod IS plasma membrane localization of Rho was previously reported in pig retina [36], in mouse and cow rods [35], and in developing rat rods (prior to OS formation) [76]. Further visualization of the Rho in IS in WT mammalian retina has likely been obscured in other reported imaging data by the overwhelming abundance of Rho in the OS and insufficient labeling of Rho in the IS. The literature does, however, contain many examples of Rho mislocalization to both the IS plasma membrane and the ONL in retinal degeneration studies using mammalian models. For example, the RP mutant Q344ter-Rho is mislocalized to the IS plasma membrane in transgenic mutant mice [11,87]. A similar widespread IS Rho

mislocalization was also seen in a RPGR mutant dog model of X-linked RP [18], in a knockout mouse for the CC-localized and JBTS-associated Tmem138 membrane protein [41], and after experimental retinal detachment in cats [88]. In each example, a perturbation to OS trafficking likely generates a Rho localization imbalance between the OS versus IS, which then highlights the IS population of Rho.

Notably, the OS peeling method we used to enable Rho immunolocalization (Fig 2A) in rod ISs may be a potential source of damage to rods; however, based on TEM analysis of IS-enriched retinas, we do not observe any obvious IS cellular damage (Fig 2D). We also did not observe any substantial mislocalization of other abundant IS proteins that we visualized in both full retinas and IS-enriched samples, including STX3 (compare Fig 4B–4E to Fig 5D–5E), PDC (Fig 4E to Fig 6E), DYNC1H1 (Fig 10G to Fig 10K), and GM130 (Fig 5B to Fig 5C). Nevertheless, in our fluorescence analyses, we only included rod ISs with intact CC/BB structures to avoid any potentially damaged cells. Furthermore, our surface labeling analysis, which was performed in full retinas that were not peeled (Fig 7), validated our finding of Rho localization at the IS plasma membrane surface and provides evidence that our Rho IS localization map is not due to tissue damage. Minor localization artifacts due to the OS peeling methods, however, cannot be completely ruled out.

Our model for IS Rho trafficking in mouse rods (depicted in S5E Fig) deviates from the highly polarized vesicular trafficking pathway in frog rods, in which Rho-containing vesicles are directed to fuse to the IS plasma membrane directly adjacent to the CC at the ciliary base (reviewed in [89]). Species differences between mouse and frog rods [52,90], which likely contribute to IS trafficking differences, include photoreceptor cell size, CC length, IS mitochondria density, and the absence of ciliary rootlet in frogs. However, we have identified commonalities between the 2 rod trafficking pathways, including roles for Rab11a, STX3, and SNAP25, which are highlighted below.

SIM localization of both Rab11a+ and GMII+ fluorescent puncta to the IS plasma membrane (Fig 10A–10F) provided further evidence that the mouse rod IS plasma membrane is an active site for protein trafficking. Based on diameter (∅) Rab11a puncta were split into 2 groups, a group with a mean ∅ approximately 125 nm and another with a mean ∅ approximately 320 nm. GMII puncta were all in the same size range (mean ∅ approximately 120 nm). Interestingly, distal IS vesicles identified in mouse cryo-EM tomograms ranged from 30 to 100 nm in diameter [9], while the majority of purified intracellular IS vesicles from frog rods were approximately 300 nm in diameter [91]; however, because of our indirect fluorescent labeling for SIM, we cannot make direct size comparisons. Next, our STORM colocalization results with Rho and Rab11a (Figs 10I, 10J, S5A, and S5B) support a potential role for this Rab GTPase in IS Rho trafficking in mouse retina as previously described [82]; however, there was a lower than expected colocalization between Rho+Rab11a, which may reflect a transient interaction between Rho and Rab11a, or a different role for Rab11a altogether in the IS. While Rab11a + fluorescent puncta may represent vesicles that are bound for SNARE-mediated vesicle fusion at the IS plasma membrane, they may alternatively mediate endocytotic events or be involved in a recently described exocytosis of approximately 150 nm microvesicles from the IS plasma membrane [92].

Any vesicular fusion events at the mouse rod IS plasma membrane would require STX3, which densely lines the IS plasma membrane of mouse rods (Fig 4B). Here, we provide further demonstration of an essential interaction between Rho and STX3 using SIM and STORM colocalization and co-IP, an interaction that was previously established in rod-specific STX3 knockout mice [55]. In frog rods, STX3 and SNAP25 also line the IS plasma membrane [54]. Furthermore, the C-terminal SNARE domain of STX3 was identified as the IS retention sequence in frog rods that kept STX3 localized to the IS and prevented OS accumulation [93].

Finally, while VAMP7 was identified as the v-SNARE in complex with STX3 and SNAP25 in frog rods [94], the functional v-SNARE in the mouse IS has not been identified [56].

We observed no consistent internal Rho localization in the ellipsoid IS or near the BB in mouse rods. Internal Rho that was sporadically localized in the ellipsoid IS may correspond to newly synthesized Rho in the ER. The ER has been previously localized in relatively close proximity to the BB in the distal mouse IS [9,15]. In the myoid IS, on the other hand, we consistently observed bright puncta of Rho that did not colocalize with the GM130+ *cis*-Golgi puncta (Fig 5C). Instead, the Rho fluorescence density was directly adjacent and possibly intercalated with the *cis*-Golgi. This finding suggests the existence of different *cis*-Golgi domains. Furthermore, the myoid rich Rho-Golgi complex in mouse rods may be analogous to Golgi exit sites in frog rods, which are also Golgi-adjacent accumulations of Rho protein in the IS [44,47].

Distal to the IS, we also observed CC membrane localized Rho using fluorescence (Figs 3B and 5A), which had previously only been observed using post-embedding immuno-EM or cryo-immuno-EM. We were also interested in visualizing Rho localization at the BB, and more specifically the DAPs, which is a critical boundary for any IS material to enter the CC. Our super-resolution localization of Rho surrounding the DAPs (Fig 8) indicates that Rho is not integrated into the DAP structure and may only interact with CEP164 and other DAP proteins indirectly. Nonetheless, the DAPs are a structural barrier between the IS and CC [30], and in a recent study, CEP164 conditional knockout mice had impaired OS disc formation defects likely due to decreased IFT protein recruitment to the BB and CC [95]. Although the role of IFT transport in rods is currently unclear [96], future efforts are needed to determine if plasma membrane–associated Rho that surrounds the DAPs is then coupled to the IFT complex along the CC.

Although we focused on the Rho localization in the IS of mouse rods, Rho was also located in the ONL (Figs 2E and 3B), where it colocalized with STX3 (Fig 4C) and was also at the surface (Fig 7D–7G); however, because the rod nuclei are so prominent in this region, a suitable nuclear membrane marker will be needed for future ONL super-resolution fluorescent localization studies. Future studies will also be needed to connect our IS localization of Rho with other previously identified mouse IS trafficking regulators, including PrBP/δ [97], IFT20 [98], myomegalin [99], and spectrin βV [100]. Nevertheless, the results presented here offer new insight into the organization of the mouse IS and may serve as the basis of a full mapping of IS trafficking processes using super-resolution fluorescence, live imaging, and other complimentary techniques.

## Materials and methods

### Animals

All WT mice were C57BL/6J between the ages of 3 weeks and 6 months. The Rho-GFP and Rho-GFP-1D4 knock-in mice were previously described [15,43] and also C57BL/6J. Mice were kept on a 12-hour light/dark cycle; however, mice used for SIM, STORM, and all western blotting experiments were dark adapted the night before experiments and killed immediately after coming into the light the next morning to normalize the light exposure and timing of all retina samples. All co-IP conditions were repeated 3 times with retina samples from 3 different mice. All SIM and STORM conditions were repeated from multiple sections from at least 2 mice. Mice from both sexes were used indiscriminately. All experimental procedures involving mice were approved by the Institutional Animal Care and Use Committee of West Virginia University (approval #2102040326).

## Antibodies and labeling reagents

The following primary antibodies were used in this study: anti-centrin (Millipore Cat# 04–1624, RRID:AB_10563501), anti-centrin-2 (BioLegend Cat# 698601, RRID:AB_2715793), anti-centrin-2 (Proteintech Cat# 15877-1-AP, RRID:AB_2077383), anti-STX3 (Millipore Cat# MAB2258, RRID:AB_1977423), anti-STX3 (Proteintech Cat# 15556-1-AP, RRID: AB_2198667), anti-Rho-C-1D4 (Millipore Cat# MAB5356, RRID:AB_2178961), anti-Rho-N-4D2 (Millipore Cat# MABN15, RRID:AB_10807045), anti-Rho-N-GTX (GeneTex Cat# GTX129910, RRID:AB_2886122), anti-ROM1 (Proteintech Cat# 21984-1-AP, RRID: AB_2878961), anti-Rab11a (Abcam Cat# ab128913, RRID:AB_11140633), anti-Tubulin/ TUBB5B (Sigma-Aldrich Cat# T7816, RRID:AB_261770), anti-PDC (custom-made rabbit polyclonal antibody, gift from Dr. Maxim Sokolov), anti-SNAP25 (BioLegend Cat# 836304, RRID:AB_2566521), anti-GM130 (BD Biosciences Cat# 610822, RRID:AB_398141), anti-mannosidase II/GMII (Abcam Cat# ab12277, RRID:AB_2139551), anti-DYNC1H1 (Proteintech, 12345-1-AP), anti-Rootletin (Millipore Cat# ABN1686, RRID:AB_2893142), and anti-ROS-GC1 (Santa Cruz Biotechnology Cat# sc-376217, RRID:AB_10991113). WGA-CF568 from Biotium (#29077) was used for surface labeling.

The following secondary antibodies were used in this study: F(ab')2-goat anti-mouse IgG Alexa 647 (Thermo Fisher Scientific Cat# A48289TR, RRID:AB_2896356), F(ab')2-goat anti-rabbit IgG Alexa 647 (Thermo Fisher Scientific Cat# A-21246, RRID:AB_2535814), F(ab') 2-goat anti-mouse IgG Alexa 488 (Thermo Fisher Scientific Cat# A-11017, RRID: AB_2534084), AffiniPure F(ab')2-goat anti-rat IgG (H+L) Alexa 488 (Jackson ImmunoResearch Labs Cat# 112-546-003, RRID:AB_2338364), F(ab')2-goat anti-mouse IgG CF568 (Biotium Cat# 20109, RRID:AB_10557119), F(ab')2-goat anti-rabbit IgG CF568 (Biotium Cat# 20099, RRID:AB_10563029), F(ab')2-goat anti-mouse IgG Alexa 555 (Thermo Fisher Scientific Cat# A-21425, RRID:AB_2535846), F(ab')2-goat anti-rabbit IgG Alexa 555 (Thermo Fisher Scientific Cat# A-21430, RRID:AB_2535851), anti-mouse IgG sdAb–FluoTag-X2 –Alexa 647 (Nanotag, #N2002-AF647-S), Nanogold-Fab' goat anti-mouse (Nanoprobes Cat# 2002, RRID: AB_2637031), and Nanogold-Fab' goat anti-rabbit (Nanoprobes Cat# 2004, RRID: AB_2631182).

To generate the GFP nanobody Alexa 647 conjugate, NbGFP-A647, the pGEX6P1-GFP plasmid (RRID:Addgene_61838) was transformed into BL21 (DE3) competent cells (Thermo Scientific) to grow recombinant GST-nanobody in Luria Broth supplemented with 50 μg/ml carbenicillin. At mid-log phase, 1,000 μM isopropyl β–d-1-thiogalactopyranoside was added to induce protein expression at 16°C. Bacterial pellets were frozen at −80°C, thawed, and incubated in the following detergent buffer: 2% Triton X-100, 0.2 mg/ml lysozyme, 20 U/ml Pierce Universal Nuclease (Thermo Scientific), 50 mM NaCl, 1 mM MgCl2, 10 mM beta-mercaptoethanol (BME), 50 mM Tris (pH 8.0), EDTA-free protease inhibitor cocktail (Bimake.com) for 0.5 to 1 h on ice. Before centrifugation, 0.5% deoxycholate and 250 mM NaCl was added to the lysates, which were then centrifuged for 30 min at $12,000 \times g$ at 4°C in a fixed-angle rotor. The cleared supernatant was loaded onto a GE ÄKTA start system fast protein liquid chromatography system using 2× 1 ml GSTrap columns (Cytiva #17528105) with a flow rate of 0.5 ml/min and 20 mM glutathione in 50 mM TRIS/HCl (pH 8.0) was used to elute the GST-nanobody. Eluates nanobody protein samples were buffer-exchanged into proteolysis buffer (25 mM HEPES, 150 mM NaCl, 1 mM DTT, 1 mM EDTA, pH 7.5) and incubated with 3 mg of Biotin HRV-3C protease (Sigma, cat# 95056) to remove the GST-tag, and then fast protein liquid chromatography on the same GSTrap columns was used again to remove the cleaved tags. Purified nanobody was buffer-exchanged into PBS (pH 8.5), validated with SDS-PAGE and Coomassie blue staining, and quantified by UV spectrophotometry.

Purified GFP nanobody was incubated with Alexa Fluor 647 NHS Ester (Thermo Fisher Scientific Cat# A20006) at a molar 1:4 ratio, and the mixture was incubated for 2 h at RT, protected from light, with occasional tapping. During this time, 2× 5 ml PD-10 G-25 desalting columns (Cytiva Cat# 17085101) were equilibrated with 1× PBS (pH 8.5). After incubation, the conjugate mix was added to 1 column and 200 μl 1× PBS (pH 8.5) fractions were collected. A NanoDrop 2000 (Thermo Scientific) was used to screen the absorbance profiles at 280 nm and 647 nm to identify the nanobody-Alexa 647 fractions. Those positive fractions were purified on the second equilibrated G-25 desalt column and screened for absorbance. The nanobody-Alexa 647 positive fractions from the second desalting purification were combined, stored at 4˚C, and used as the NbGFP-A647 reagent for immunolabeling.

## Retinal immunofluorescence

For immunofluorescence and confocal microscopy, mouse eyes were enucleated and a small puncture was made in the cornea and eyes was subsequently fixed for 15 min in 4% PFA at RT. The cornea and lens were then completely removed to form an eye cups, which were further fixed in 4% PFA for 45 min at RT. Following incubation in 30% sucrose for cryopreservation, eye cups were transferred to a 1:1 mix of 30% sucrose and optimal cutting temperature (OCT) medium for additional cryopreservation. Eye cups were frozen in cryomolds, and 16 μm sections were collected on either a Leica Cryostat CM1850 or a Medical Equipment Source 1000+ Cryostat and collected on Superfrost slides (VWR, Cat# 48311–703). For immunolabeling, sections were quenched with 100 mM glycine diluted in 1× PBS for 10 min at RT. Sections were then incubated with blocking solution: 10% normal goat serum (NGS) (Fitzgerald), 0.3% Triton X-100, 0.02% sodium azide, diluted in 1× PBS, for 1 h at RT. Between 1 μg to 5 μg of primary antibodies were diluted in 200 μl the same block solution and added to the sections for overnight probing at 4˚C in a humidified chamber. The next day, sections were washed and probed with 1 μg of fluorescent secondary antibodies diluted in 200 μl of the same blocking solution for 1.5 h at RT. After washing, sections were counterstained with 0.2 μg/ml 4′,6-diamidino-2-phenylindole (DAPI) (Thermo Fisher Cat# 62248) diluted in 1× PBS for 15 to 30 min at RT. Sections were postfixed in 4% PFA for 5 min and mounted with ProLong Glass Antifade Mountant (Thermo Fisher Scientific Cat# P36980). Confocal scanning was performed at RT on either a Nikon C2 confocal microscope equipped with photomultiplier detectors using 40× Plan Fluor, NA 1.3 and 100× Plan Apo, NA 1.45 oil immersion objectives, or a Nikon A1R confocal microscope equipped with GaAsP and photomultiplier detectors using a 40× Plan Fluor, NA 1.3 objective. Confocal images were acquired using NIS-Elements software and processed using Fiji/ImageJ [101].

For SIM and STORM immunofluorescence, mouse retinas were dissected in ice-cold buffered Ames media (Sigma, Cat# A1420) and either lightly fixed in 4% PFA diluted in Ames' for 5 min on ice (for whole retina samples) or immediately underwent retinal peeling for IS enrichment. The peeling procedure is adapted from [65,66]. Dissected retinas for peeling were first bisected and then trimmed into rectangular retinal slices. Then, 5.5 cm filter paper (VWR, Cat# 28310–015) was cut into small squares, a piece of filter paper was added to dish of Ames' media containing the retinal slices, and the slices were oriented with the OS facing the filter paper. The filter paper with retina slice attached was carefully lifted out of solution and blotted dry filter side down on a paper towel before being transferred back into the Ames' media, and this blotting procedure was repeated 4 times. After blotting, the retina was carefully peeled away from the filter paper using forceps. This process constituted a single peel. In total, the retina slices were peeled 8 times (for IS-enriched retinas), a procedure that takes approximately 10 min, before immediately being lightly fixed in 4% PFA diluted in Ames' for 5 min on ice.

After fixation, retina samples (either whole retinas or IS-enriched retinas) were quenched in 100 mM glycine at 4°C and incubated in 1 mL SUPER block solution: 15% NGS, 5% bovine serum albumin (BSA) (Sigma, Cat# B6917) + 0.5% BSA-c (Aurion, VWR, Cat# 25557) + 2% fish skin gelatin (Sigma, Cat# G7041) + 0.05% saponin (Thermo Fisher, Cat# A1882022) + 1× protease inhibitor cocktail (GenDepot, Cat# P3100-005), in half dram vials (Electron Microscopy Sciences, Cat# 72630–05) or low-adhesion microcentrifuge tubes (VWR, Cat# 49003–230) for 3 h at 4°C. Between 1 µg to 5 µg of primary antibodies or 6 µg NbGFP-A647 was added to 1 ml of the blocking solution for probing at 4°C for 24 h with mild agitation. The next day, a second dose of Rho antibodies or NbGFP-A647 (same amounts as the day prior) was added to improve retinal labeling penetration, and the retinas were probed at 4°C for an additional 48 h with mild agitation (for 72 h total of primary immunolabeling). Retinas were washed 6 times for 10 min each in 2% NGS diluted in Ames' prior to probing with 4 µg to 8 µg of secondary antibodies diluted in 1 ml of 2% NGS in Ames' + 1× protease inhibitor cocktail at 4°C for 12 to 16 h with mild agitation. Retinas were then washed 6 times, 5 min each in 2% NGS diluted in Ames' and fixed in 2% PFA + 0.5% glutaraldehyde diluted in 1× PBS for 30 min at 4°C with mild agitation. Retinas were then dehydrated in an ethanol series with the following steps of pure ethanol diluted in water: 50%, 70%, 90%, 100%, 100%; each step for 15 min in half dram vials on an RT roller. Dehydrated retinas were then embedding in Ultra Bed Low Viscosity Epoxy resin (Electron Microscopy Sciences, EMS Cat# 14310) in the following series of steps: 1:3 resin to 100% ethanol for 2 h rolling at RT; 1:1 resin to 100% ethanol for 2 h rolling at RT; 3:1 resin to 100% overnight (approximately 16 h) rolling at RT; 2 steps of full resin (no ethanol), 2 h each, rolling at RT. Resin-embedded retina samples were mounted in molds and cured in baking oven set to 65°C for 24 h. A Leica UCT ultramicrotome and glass knives were used to make 0.5 µm to 1 µm thin retinal cross sections.

For surface labeling experiments, dissected retinas were immediately transferred, unfixed, into SUPER block solution with no saponin for a 3-h blocking incubation step prior to overnight primary staining, again with no saponin (both at 4°C with mild agitation). For these experiments, 5 µg of WGA-CF548 was used and added during the primary staining step. The next day, surface labeled retinas were washed 6 times for 10 min each in 2% NGS diluted in Ames' prior to probing with 4 µg to 8 µg of secondary antibodies diluted in 1 ml of 2% NGS in Ames' + 1× protease inhibitor cocktail + 0.05% saponin at 4°C for 3 h with mild agitation. Retinas were then washed 6 times, 5 min each in 2% NGS diluted in Ames', and then fixed in 4% PFA for 20 min at RT with mild agitation. Retinas were then dehydrated and resin embedded for thin sectioning as described above.

For SIM, thin retina resin sections were collected onto #1.5 coverslips (VWR, Cat# 1152222), which were then mounted onto a plain glass slides with ProLong Glass and cured at RT for at least 2 full days protected from light. For STORM, thin retina resin sections were collected in 35 mm glass-bottom dishes (MatTek Life Sciences, Cat# P35G-1.5-14-C), chemically etched in a mild sodium ethoxide solution (approximately 1% diluted in pure ethanol for 0.5 to 1.5 h), as previously described [57]. Prior to STORM imaging, etched sections were mounted in a STORM imaging buffer adapted from [58]: 50 mM Tris (pH 8.0), 10 mM NaCl, 10 mM sodium sulfite, 10% glucose, 40 mM cysteamine hydrochloride (MEA, Chem Impex/VWR, Cat# 102574–806), 143 mM BME, and 1 mM cyclooctatetraene (Sigma Cat# 138924), under a #1.5 glass coverslip that was sealed with quick-set epoxy resin (Devcon).

An alternative protocol was used for SIM imaging of GFP fluorescence in Rho-GFP/+ and Rho-GFP-1D4/+ retinal cryosections. Enucleated eyes were immersion fixed in 4% PFA diluted in 1× PBS for 1 h at RT and incubated in a hydrogel solution: 4% acrylamide (Sigma, Cat# A4058), 2% bis-acrylamide (Alfa Aesar Cat# J63265), 2.5% VA-004 (TCI America), overnight at 4°C. The hydrogel was polymerized for 5 min at 37°C and degassed for 15 min at RT.

Hydrogels were cryoprotected with sucrose, embedded in OCT, and frozen. Then, 2 μm cryosections were collected on Superfrost slides, rinsed in 1× PBS, mounted in ProLong Glass Antifade Mountant, and cured for 2 days at RT before SIM.

## SIM

A Nikon N-SIM E microscope system equipped with a Hamamatsu Orca-Flash 4.0 camera was used for imaging at RT with a SR HP Apochromat TIRF 100X, NA 1.49 oil immersion objective. NIS-Elements Ar software was used for image acquisition, and Z-projections were collected from regions of interest (containing 5 to 15 slices), using a 0.2-μm Z-section thickness. Samples were imaged using 488 nm, 561 nm, and 647 nm lasers with 15 grating pattern images. SIM reconstructions were performed in NIS-Elements software and assessed for quality based on the fast Fourier transform (FFT) images from the reconstructions. 3D deconvolution was also performed in Nikon NIS-Elements software using Automatic deconvolution mode.

## STORM

All STORM acquisitions were performed at RT on a Nikon N-STORM 5.0 system equipped with an Andor iXON Ultra DU-897U ENCCD camera with a SR HP Apochromat TIRF (total internal reflection fluorescence) 100X, NA 1.49 oil immersion objective. The system also features a piezo Z stage,100 mW 405 nm, 488 nm, and 561 nm laser lines and a 125 mW 647 nm laser line, as well as a Lumencor SOLA Light Engine epifluorescence light source. For STORM acquisition, Nikon NIS-Elements software was used, and sections and imaging fields of interest were located using widefield epifluorescence. All STORM was performed in 2D mode using 40 μm × 40 μm field of view dimensions. At low laser power, 561 nm and 647 nm lasers were used to identify spots with multiple bright CC, and then pre-STORM images were acquired using epifluorescence and lower laser power. GFP and Alexa 488 fluorescence were photobleached using 488 nm laser at maximum power prior to using the 561 nm and 647 nm lasers at 90% to 100% max power to initiate CF568 and Alexa 647 photoswitching. For optimal photoswitching, 2× laser magnification was used, and all STORM acquisitions were performed near the TIRF critical angle; the TIRF angle and direction values were adjusted for each sample to optimize signal-to-noise in each imaging field. STORM data were acquired at approximately 33 frames per second, and 40,000 frames were collected per channel for each STORM acquisition. For all 2-channel STORM 561 nm and 647 nm frames were collected sequentially. STORM analysis was performed using Nikon NIS-Elements software with analysis parameters matching the "single molecule" settings from [57] to detect only bright, individual photoswitching events, as follows: minimum point spread function (PSF) height: 1,000, maximum PSF height: 65,000, minimum PSF width: 200 nm, maximum PSF width: 400 nm, initial fit width: 300 nm, max axial ratio: 1.15, max displacement: 1 pixel. The resulting STORM reconstructions were populated as single molecule events with localization errors <20 nm. Chromatic aberration between channels was corrected using an X-Y bead warp calibration, and drift correction was performed using an autocorrelation algorithm.

## TEM

IS-enriched or full (unpeeled control) retinas for TEM were immediately fixed in 2% PFA + 2% glutaraldehyde + 4.5 mM $CaCl_2$ in 50 mM MOPS buffer (pH 7.4) for 4 to 5 h at 4°C on a roller. Retinas were then incubated in half dram vials containing 1% tannic acid (EMS, Cat# 21700) + 0.5% saponin diluted in 0.1 M HEPES (pH 7.5) for 1 h at RT, then rinsed 4 times with water before incubation with 1% uranyl acetate (EMS, Cat# 22400) diluted in 0.1 M

maleate buffer (pH 6.0) for 1 h at RT. Retinas were then rinsed 4 times in water and then dehydrated in an ethanol series with the following steps of pure ethanol diluted in water: 50%, 70%, 90%, 100%, 100%; each step was for 15 min in half dram vials on an RT roller. Dehydrated retinas were then embedded in Ultra Bed Low resin and cured in resin molds as described for SIM/STORM retinas above.

For ultrathin sectioning, 70 nm ultramicrotome sections were cut using a Diatome Ultra 45˚ diamond knife and collected onto square 100 mesh copper grids (EMS, Cat# G100-Cu). Grids were poststained in 1.2% uranyl acetate diluted in water for 4 min, rinsed 6 times in water, and allowed to completely dry before staining with either a Sato's Lead solution (1% lead acetate (EMS, Cat# 17600), 1% lead citrate (EMS, Cat# 17800), 1% lead nitrate (EMS, Cat# 17900), 2% sodium citrate diluted (EMS, Cat# 21140) in water), or a lead citrate solution (EMS, #22410) for 4 min. Grids were then rinsed in water and dried overnight. Grids were imaged using a JEOL JEM 1010 transmission electron microscope equipped with a side-mounted Hamamatsu Orca-HR digital camera at 25,000× magnification.

### Immuno-EM

Retinas were dissected, IS-enriched, and immunolabeled as described in the SIM/STORM section except the prefixation solution was 4% PFA + 2.5% glutaraldehyde in Ames', nanogold-conjugated secondaries were used (5 to 7.5 μg), and the postfixation solution was 2% PFA + 2% glutaraldehyde + 4.5 mM $CaCl_2$ in 50 mM MOPS buffer (pH 7.4). Retinas were then rinsed with water and enhanced using HQ Silver Kit (Nanoprobes, Cat# 2012) reagents in half dram vials for 4 min at RT with agitation. Enhanced retinas were then immediately rinsed with water, incubated in 1% tannic acid + 0.5% glutaraldehyde in 0.1 M HEPES (pH 7.5) for 1 h on an RT roller, rinsed with water, incubated in 1% uranyl acetate in 0.1 M maleate buffer (pH 6.0) for 1 h on an RT roller, and rinsed a final time with water. Retinas were then ethanol dehydrated and resin embedded in Ultra Bed resin as outlined in the TEM section. For ultrathin sectioning, 100 nm ultramicrotome sections were cut using a Diatome Ultra 45˚ diamond knife and collected onto square 100 mesh copper grids. Grids were imaged as for TEM.

### Image processing and spatial analysis

Fiji/ImageJ was used to generate confocal and SIM z-stack projections, to adjust whole image brightness and contrast adjustments for clarity, and to generate row average intensity profiles (using the "Plot Profile" function). For SIM fluorescent puncta analyses, puncta membrane localization was scored using SIM z-projection images. Puncta diameters were acquired by boxing single puncta in Fiji/ImageJ, acquiring row average intensity profiles, and using the "Add Fit. . ." function to fit Gaussian functions to the profiles, which were exported to a custom code in Mathematica v13.1 (Wolfram) to determine full width half maximum (FWHM) diameter values.

STORM reconstruction data were processed in NIS Elements Ar v5.30.05 (Nikon) using the N-STORM Analysis modules. Rod IS hulls were manually drawn on STORM reconstructions using the "Draw Polygonal ROI. . ." tool, and molecule coordinates within IS hulls were exported for further processing. Widefield images were superimposed onto STORM reconstructions and exported as image files to be adjusted, scaled, and cropped for visualization in Fiji/ImageJ. STORM spatial analysis was performed using custom code in Mathematica v13.1.0.0 (Wolfram). Molecule coordinates corresponding to single rod IS hulls were imported and plotted using the ListPlot function, and the IS hull was defined using the ConcaveHull-Mesh function on the STX3 coordinates (or Rho-N-4D2 coordinates for Fig 7 data). Random points were generated using the RandomReal function, with points outside the IS hull

discarded. Nearest distance-to-hull measurements were calculated using the SignedRegionDistance function and graphed as frequency plots or CDF plots using the PDF or CDF functions, respectively. Finally, $\leq 0.1$ µm, $\leq 0.2$ µm, or $\leq 0.3$ µm frequency values were calculated as the number of molecules with nearest distance-to-hull within the indicated range, divided by the total number of molecules.

The Mosaic IA Fiji/ImageJ plugin [74] was used to perform the STORM spatial interaction analysis to quantify STORM colocalization. Coordinates for each channel from an IS hull were uploaded (with the STX3 channel serving as the reference channel). The following settings were used: Grid spacing = 0.03, Kernel wt(q) = 0.001, Kernel wt(p) = based on the estimate provide by the program. The parameterized "Hernquist potential" was used to calculate the interaction strength values. Rab11a clusters were visualized in STORM reconstructions in NIS-Elements Ar using as Voronoi tessellations using the Molecule Analysis feature within the N-STORM Analysis program; Voronoi parameters were set to cluster 3 or more molecules in a 10-nm maximum distance. Tessellations were visualized using the Render Clusters function and superimposed onto STORM reconstruction images in NIS-Elements.

## Western blotting

For retinal lysate western blotting, mouse retinas were dissected, and residual retinal pigment epithelium and ciliary body material were removed. Single retinas were then frozen individually on dry ice for at least 10 min before adding 200 µl of urea sample buffer: 6 M urea, 140 mM SDS, approximately 0.03% bromophenol blue diluted in 0.125 M Tris (pH 6.8) and supplemented with 360 mM BME. Samples were sonicated to lyse the retinas. Lysates were centrifuged 1 to 5 min, and supernatants were collected. Samples were loaded onto Novex WedgeWell 10% to 20% Tris-Glycine, 0.1 mm, MiniProtein Gels (Invitrogen, Cat# XP10202BOX) along with the Precision Plus Dual Color ladder (Bio-Rad, Cat# 1610374) in Tris-Glycine-SDS running buffer (Bio-Rad, Cat# 1610772) for SDS-PAGE. Gels were transferred onto Immobilon-FL Transfer Membrane PVDF (pore size: 0.45 µm) (LI-COR Cat# 92760001) in Tris-Glycine Transfer Buffer (Bio-Rad Cat# 1610771) + 10% methanol. Membranes were subsequently blocked using Intercept Blocking Buffer (LI-COR, Cat# 927–6000) for 1 h and washed before primary antibody staining in 1× PBS + 0.1 Tween-20 (PBS-T) (antibodies were used at 1:500 to 1:5,000 dilutions) for 1 h. Membranes were washed in PBS-T 3 times, 5 min each before secondary staining with Alexa 647 secondary antibodies (1:50,000 each) diluted in PBS-T for 1 h. Secondary antibodies used were F(ab')2-goat anti-mouse IgG Alexa 647 and F(ab')2-goat anti-rabbit IgG Alexa 647. Membranes were washed then imaged for fluorescence on an Amersham Typhoon scanner (GE) using both Cy5 and IR-Short fluorescence filter sets; the IR-short channel was optimal for imaging the protein ladder.

## Co-immunoprecipitation

Retinas were dissected and IS-enriched as described above before each individual retina was frozen on dry ice. For lysis 200 µl T-PER lysis buffer (Thermo Fisher Scientific, Cat# 78510) was added to frozen retinas prior to sonication. Retinal lysates were centrifuged, soluble supernatants were collected, and 4 µl supernatant was saved as "input" fractions for SDS-PAGE (corresponding to 2% of a lysed retina). Then, 20 µl of either anti-GFP Trap magnetic beads (Chromotek, Cat# gtma) or 1D4-Ab-conjugated sepharose beads (approximately 30% slurry, gift from Dr. Theodore Wensel) was added directly to remaining lysates and incubated overnight at 4°C with agitation. Following overnight incubation, samples were pelleted, and 4 µl supernatant was removed as the "unbound" fraction for SDS-PAGE (also corresponding to 2% of a lysed retina). For magnetic beads, a magnetic stand was used to pellet beads for washes.

For agarose beads, samples were centrifuged to pellet the beads. All beads were washed a total of 4 times for 3 min each in PBS-T at RT. After washing, the "bound" sample was eluted from beads using 25 µl urea sample buffer supplemented with 360 mM BME. Half of the bound eluate sample was loaded alongside the corresponding input and unbound samples for SDS-PAGE in Novex WedgeWell 10% to 20% Tris-Glycine, 0.1 mm, MiniProtein Gels (Invitrogen) with Tris-Glycine-SDS running buffer (BioRad). Western blot transfer, immunolabeling, and scanning steps were performed as described in the western blotting section.

### Deglycosylation assay

Full mouse retinas were dissected as described above. After freezing, 200 µl of RIPA lysis buffer (Alfa Aesar, Cat# J63306) supplemented with 1× protease inhibitor cocktail (GenDepot, Cat# P3100-005) was added directly to a frozen retina, which was sonicated and centrifuged as described above. Supernatant concentrations were measured using a NanoDrop 2000 (Thermo Fisher Scientific). Using Protein Deglycosylation Mix II (NEB Cat# P6044S), 100 µg of lysates were deglycosylated. In short, lysates were incubated with 1× Deglycosylation Buffer 2 for 10 min at 37˚C, then 1× Deglycosylation Mix II (containing PNGase F) was subsequently added, and samples were incubated for 1 h at 37˚C. The reaction was terminated by cooling lysates on ice for 5 to 10 min. Lysates were then further diluted to 50 µg in urea sample buffer supplemented with BME for western blotting.

### Statistical analysis

Mann–Whitney $U$ tests were performed in Microsoft Excel by ranking the values from each condition and calculating the $U$ test and z test statistics (to derive $P$ values via hypothesis testing). One-sample $t$ tests for comparisons to the hypothetical mean value of 1 were performed using GraphPad QuickCalcs (https://www.graphpad.com/quickcalcs/). One-way ANOVA analysis (Anova: Single Factor) was performed in Microsoft Excel using the Analysis ToolPak Excel Add-in. Data were visualized for comparison as violin plots using PlotsOfData [102].

### Supporting information

**S1 Fig.** (**A**, **B**) Western blot test of NbGFP-A647 immunolabeling specificity. (**A**) WT and Rho-GFP/+ retinal lysates used as either 0.5% or 2% total volume of 1 mouse retina. In Rho-GFP/+ lysates, a prominent and specific NbGFP+ band was found approximately 60 kDa corresponding to monomeric Rho-GFP protein. Both WT and Rho-GFP/+ lysates contained Rho-C-1D4-positive bands corresponding to endogenous mouse Rho protein. Mouse Rho protein levels are apparently lower in Rho-GFP/+ lysates—corresponding to the one WT *Rho* allele in these mice. In addition, Rho-C-1D4 antibody does not immunolabel Rho-GFP protein from Rho-GFP/+ lysates. IB, immunoblot condition. (**B**) WT and Rho-GFP-1D4/+ retinal lysates used as 2% total volume of 1 mouse retina. In Rho-GFP-1D4/+ lysates specific NbGFP-A647+ and Rho-C-1D4+ bands are found approximately 60 kDa corresponding to monomeric Rho-GFP-1D4 protein. As is (**A**), endogenous mouse Rho protein levels are lower in Rho-GFP-1D4/+ lysates. For all western blots, molecular weight marker sizes are indicated in kDa. (**C**) To test NbGFP-A647 immunolabeling specificity with immunofluorescence, 10 µm eye cup cryosections from Rho-GFP/+ mice (age P71) and from WT mice (age P60) were immunolabeled with NbGFP-A647 and counterstained with DAPI. NbGFP-647 specifically labels the OSs in the Rho-GFP/+ section (magenta) and overlaps with GFP (green). No detectable NbGFP-A647 fluorescence was detected in the WT sections. Scale bar values match adjacent panels when not given. (**D**) Rho-GFP/+ or (**E**) Rho-GFP-1D4/+ adult retinal lysates treated with either Protein Deglycosylation Mix II from NEB (containing PNGase F) or buffer

only. After treatment, 5 µg of total protein was loaded for SDS-PAGE, transfer, and primary antibody labeling. A shift to lower molecular weight was observed for all Rho bands, including Rho-GFP bands from Rho-GFP/+ treated lysates probed with NbGFP-A647. In (**D**), Rho-GFP was also detected with Rho-N-4D2 immunolabeling; monomeric Rho-GFP bands are indicated with green arrows on this blot. In (**E**), Rho-GFP-1D4 was detected with Rho-C-1D4 immunolabeling and monomeric Rho-GFP-1D4 bands are indicated with green arrows. NbGFP-A647, GFP nanobody Alexa 647 conjugate; OS, outer segment; Rho, rhodopsin; WT, wild-type.
(PDF)

**S2 Fig.** (**A**) TEM images of Rho-GFP/+ retina slices that were either unpeeled, peeled 4 times, or peeled 8 times (the IS-enriched condition). Images were pseudocolored to point out key rod structures as follows: OS = yellow, IS = magenta, CCs/BBs = blue. Scale bar values match adjacent panels when not labeled. (**B**) Alternate TEM single rod examples from Rho-GFP/+ IS-enriched retinas. The IS plasma membrane is annotated with magenta arrows. (**C**) SIM images from IS-enriched Rho-GFP/+ retinas immunolabeled for NbGFP-A647 (magenta), STX3 (cyan), and centrin (yellow). To demonstrate Rho-GFP colocalization with STX3 at the IS plasma membrane, row average intensity plots are shown for portions of the IS from 2 different magnified single rod examples marked with a dashed line. (**D**, **E**) Additional single rod SIM z-projection images of WT IS-enriched retina sections immunolabeled with either (**D**) Rho-C-1D4 (magenta) or (**E**) Rho-N-4D2 (magenta); both co-immunolabeled with STX3 (cyan) and centrin-2 (yellow) antibodies. White arrows indicate Rho fluorescence that is colocalized with STX3 at the plasma membrane. Numerical values corresponding to all graphical data are provided in Table F in S1 Data. BB, basal body; CC, connecting cilium; IS, inner segment; NbGFP-A647, GFP nanobody Alexa 647 conjugate; OS, outer segment; Rho, rhodopsin; SIM, structured illumination microscopy; STX3, syntaxin 3; TEM, transmission electron microscopy; WT, wild-type.
(PDF)

**S3 Fig.** Alternate single rod STORM reconstruction examples from (**A**) Rho-GFP staining conditions colabeled with STX3 (cyan) and centrin-2 yellow. Each example features plots of STORM molecule coordinates within the STX3+ IS hull for each channel and an adjacent plot with randomly plotted molecules within the IS hull (orange), as well as frequency and CDF graphs for distance to hull measurements from the plotted STORM coordinates. Molecule counts: (**Ai**) Rho-GFP $n$ = 9,634, STX3 $n$ = 7,643, Random $n$ = 9,634; (**Aii**) Rho-GFP $n$ = 4,298, STX3 $n$ = 14,385, Random $n$ = 4,298; (**Aiii**) Rho-GFP $n$ = 4,709, STX3 $n$ = 10,385, Random $n$ = 4,709. (**B**) STORM reconstructions of IS-enriched Rho-GFP-1D4/+ retina sections immunolabeled with NbGFP-A647 (cyan) and STX3 (cyan) and centrin-2 (yellow) antibodies. OSs are indicated. The centrin-2+ widefield images are superimposed on STORM reconstruction images. In single rod examples, the IS region is indicated, and the IS hull is outlined in cyan in a duplicate image. For each example Rho-GFP-1D4 and STX3 STORM molecule coordinates within the IS hull are plotted (top example: Rho-GFP-1D4 molecules = 8,135, STX3 molecules = 24,826; bottom example: Rho-GFP-1D4 molecules = 3,326, STX3 molecules = 17,765). In the adjacent plot, a random distribution of coordinates within the IS hull matching the number of Rho-GFP-1D4 molecules (4,239) are plotted in orange. Nearest distance-to-hull measurements for Rho-GFP-1D4, STX3, and random molecules are plotted in a frequency and CDF graphs. Colors in the graphs match the molecule plots. (**C**-**F**) Alternate single rod STORM reconstruction examples from (**C**) Rho-C-1D4, (**D**) Rho-N-4D2, (**E**) SNAP25, and (**F**) PDC conditions all colabeled with STX3 (cyan) and centrin-2 (yellow) antibodies. Molecule counts (**Ci**) Rho-C-1D4 $n$ = 1,255, STX3 $n$ = 60,036, Random $n$ = 1,255; (**Cii**)

Rho-C-1D4 $n$ = 9,586, STX3 $n$ = 37,569, Random $n$ = 9,586; (**Ciii**) Rho-C-1D4 $n$ = 1,152, STX3 $n$ = 66,161, Random $n$ = 1,152; (**Di**) Rho-N-4D2 $n$ = 4,965; STX3 $n$ = 67,836; Random $n$ = 4,965; (**Dii**) Rho-N-4D2 $n$ = 4,975; STX3 $n$ = 13,387; Random $n$ = 4,975; (**E**) SNAP25 $n$ = 7,697; STX3 $n$ = 25,196; Random $n$ = 7,697; (**F**) PDC $n$ = 10,920; STX3 $n$ = 18,619, Random $n$ = 10,920. Black arrows = Rho STORM molecules located at the STX3+ IS hull in rod examples where the majority of Rho molecules are internal (**Aii**, **Cii**, **Dii**). Numerical values corresponding to all graphical data are provided in Table G in S1 Data. CDF, cumulative distribution function; IS, inner segment; NbGFP-A647, GFP nanobody Alexa 647 conjugate; OS, outer segment; PDC, phosducin; Rho, rhodopsin; STORM, stochastic optical reconstruction microscopy; STX3, syntaxin 3.
(PDF)

**S4 Fig.** (**A**, **B**) Violin plot graphs of STORM distance to hull normalized frequency values within (**A**) 0.1 μm and (**B**) 0.3 μm. $N$ values are the same as in Fig 6F. Comparisons were tested for statistical significance using the Mann–Whitney $U$ test. (**A**) PDC vs. Rho-GFP **$P$ value = 0.0011; PDC vs. Rho-GFP-1D4 ***$P$ value < 0.00001; PDC vs. 1D4 ***$P$ = 0.0003; PDC vs. 4D2 *$P$ = 0.0172. (**B**) PDC vs. Rho-GFP ***$P$ value = 0.0003; PDC vs. Rho-GFP-1D4 ***$P$ value < 0.00001; PDC vs. 1D4 **$P$ = 0.0033. (**C**, **D**) Immunogold localization of STX3 and Rho in mouse rods. (**C**) Single rod IS electron micrograph examples from a WT mouse retinas immunolabeled with STX3 antibody and nanogold secondary antibody. In a threshold image showing only the STX3+ immunogold particles, the approximate location of the IS plasma membrane is outlined in cyan. The outline is also continuous with the CC membrane. (**D**) Electron micrograph examples of rod ISs from IS-enriched WT mouse retinas immunolabeled with Rho-C-1D4. The IS plasma membrane is outlined in cyan in the threshold image. Some non-punctate staining from the BB and CC axoneme is present in the threshold images. mito = mitochondria. Numerical values corresponding to all graphical data are provided in Table H in S1 Data. BB, basal body; CC, connecting cilium; IS, inner segment; PDC, phosducin; Rho, rhodopsin; STORM, stochastic optical reconstruction microscopy; STX3, syntaxin 3; WT, wild-type.
(PDF)

**S5 Fig.** (**A**) Alternate STORM single rod examples from Rho-GFP/+ IS-enriched retinas immunolabeled with NbGFP-A647 (magenta), and centrin and STX3 antibodies (combined, yellow), and Rab11a. (**B**) STORM examples for Rho-GFP-1D4/+ IS-enriched retinas immunolabeled with NbGFP-A647 (magenta), and centrin and STX3 antibodies (combined, yellow), and Rab11a. Rab11a+ clusters identified with Voronoi tessellation are in white. (**C**, **D**) STORM examples for Rho-GFP-1D4/+ IS-enriched retinas immunolabeled with NbGFP-A647 (magenta), and centrin and STX3 antibodies (combined, yellow), and either (**C**) DYNC1H1 or (**D**) Rootletin antibody. (**E**) Diagram of the mouse rod IS architecture and the localization pattern of Rho. IS, inner segment; NbGFP-A647, GFP nanobody Alexa 647 conjugate; Rho, rhodopsin; STORM, stochastic optical reconstruction microscopy; STX3, syntaxin 3.
(PDF)

**S1 Data. Tables containing numerical values corresponding to all graphical data that are presented in the figures.**
(XLSX)

**S1 Raw Images. Source Data for Figs 2, 8, and S1.**
(PDF)

## Acknowledgments

The authors thank Dr. Paolo Fagone and WVU Department of Biochemistry and Molecular Medicine Viral Core Facility for assistance with GFP nanobody purification. The authors thank Drs. Abigail Moye, Ezequiel Salido, Maxim Sokolov, and Theodore Wensel, as well as Robichaux lab members, for discussion and feedback during manuscript preparation. The authors declare no competing financial interests.

Confocal and SIM imaging experiments were performed in the West Virginia University Microscope Imaging Facility, which has been supported by the WVU Cancer Institute and National Institute of Health research grants P20RR016440 and P30RR032138/P30GM103488. The Nikon A1R-SIM is supported with funding from U54GM104942 and P20GM103434.

## Author Contributions

**Conceptualization:** Kristen N. Haggerty, Shannon C. Eshelman, Michael A. Robichaux.

**Funding acquisition:** Michael A. Robichaux.

**Investigation:** Kristen N. Haggerty, Shannon C. Eshelman, Lauren A. Sexton, Emmanuel Frimpong, Leah M. Rogers, Michael A. Robichaux.

**Methodology:** Kristen N. Haggerty.

**Project administration:** Michael A. Robichaux.

**Resources:** Michael A. Robichaux.

**Software:** Melina A. Agosto, Michael A. Robichaux.

**Supervision:** Michael A. Robichaux.

**Visualization:** Kristen N. Haggerty, Melina A. Agosto, Michael A. Robichaux.

**Writing – original draft:** Michael A. Robichaux.

**Writing – review & editing:** Kristen N. Haggerty, Shannon C. Eshelman, Melina A. Agosto, Michael A. Robichaux.

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
