## [Editor Report · Decision Letter 0]

25 May 2023

Dear Dr Robichaux, 

Thank you for submitting your manuscript entitled "Mapping rhodopsin trafficking in rod photoreceptors with quantitative super-resolution microscopy" for consideration as a Research Article by PLOS Biology.

Your manuscript has now been evaluated by the PLOS Biology editorial staff as well as by an academic editor with relevant expertise and I am writing to let you know that we would like to send your submission out for external peer review.

Once your full submission is complete, your paper will undergo a series of checks in preparation for peer review. After your manuscript has passed the checks it will be sent out for review. To provide the metadata for your submission, please Login to Editorial Manager (https://www.editorialmanager.com/pbiology) within two working days, i.e. by May 29 2023 11:59PM.

Kind regards,

Ines

--

Ines Alvarez-Garcia, PhD

Senior Editor

PLOS Biology

---

## [Decision Letter · Decision Letter 1]

26 Jul 2023

Dear Dr Robichaux,

Thank you for your patience while your manuscript "Mapping rhodopsin trafficking in rod photoreceptors with quantitative super-resolution microscopy" was peer-reviewed at PLOS Biology. It has now been evaluated by the PLOS Biology editors, an Academic Editor with relevant expertise, and by several independent reviewers. 

In light of the reviews, which you will find at the end of this email, we would like to invite you to revise the work to thoroughly address the reviewers' reports.

As you will see below, the reviewers find the results of your study interesting. However, Reviewer 1 and Reviewer 2 raise several concerns that should be addressed before we consider the manuscript for publication. In particular, both reviewers think that your approach could potentially produce artificial rhodopsin mislocalization, so they propose some experiments to confirm the conclusions. 

Given the extent of revision needed, we cannot make a decision about publication until we have seen the revised manuscript and your response to the reviewers' comments. Your revised manuscript is likely to be sent for further evaluation by all or a subset of the reviewers.

**IMPORTANT - SUBMITTING YOUR REVISION**

*Re-submission Checklist*

*Published Peer Review*

*PLOS Data Policy*

*Blot and Gel Data Policy*

Sincerely,

Christian (on behalf of Ines)

Ines Alvarez-Garcia, PhD

Senior Editor

PLOS Biology

REVIEWS:

Reviewer's Responses to Questions

Reviewer #1: Rhodopsin is synthesized in the inner segment and traffics to the outer segment in the highly compartmentalized photoreceptor cell. This process is important for the maintenance of photoreceptor health and function. The manuscript by Haggerty et al. employed super-resolution microscopy that revealed the localization of rhodopsin in mouse rod inner segments. The images showed localization of rhodopsin and molecules involved in rhodopsin trafficking in remarkable detail, adding to our understanding of this complicated process. A major concern is that much of the work relies on the removal of the outer segment compartments in live tissue. This process could potentially produce artificial rhodopsin mislocalization. This concern is underscored by the common observation that rhodopsin tends to mislocalize in stressed, degenerating retina. Providing an experiment as well as thoughtful discussion to refute this possibility would greatly strengthen the manuscript. 

Minor comments:

1. Some paragraphs in the introduction read like general reviews and can be condensed. How the content of these paragraphs directly relates to the question being addressed by the manuscript is often not clear. A concluding sentence at each paragraph linking the cited studies to questions being addressed by the current study would be helpful.

2. Line 21: efficient long-term phototransduction. Consider rephrasing.

3. Line 56: CC is a thin ciliary bridge that spans 1.1 um. It would be more clear to define the diameter as well as length.

4. Line 174: Rho-C-1D4 labeling sentence is a repeat of line 170.

5. What is the conclusion from Fig. 1F data? There is no rhodopsin in the inner segment plasma membrane? What is the interpretation between the conflicting results in Fig. 1 C, D and F?

6. Line 294: Fig. 4D, E should be Fig. 5D, E.

7. The relatively poor colocalization between rhodopsin and rab11a in the inner segment is somewhat surprising. Do the authors expect this low degree of colocalization? Caption for Fig. 9 should reflect this partial colocalization.

Reviewer #2: Haggarty et al

This study applies quantitative super-resolution microscopy (SIM and STORM) to uncover localization of rhodopsin in rod photoreceptors and determine details about rhodopsin trafficking in the mouse rod IS. The authors use WT and a hRho-GFP/+ knockin mouse model in which the mouse rhodopsin gene was replaced by the human gene encoding a hRho-GFP fusion protein. Key results are that in confocal images of Rho-GFP/+ and Rho-GFP-1D4/+ retinas, GFP fluorescence was confined to the OS layer. Using SIM, a faint GFP signal was visualized in the IS layer from both knockin mice. Fluorescence with a GFP-specific nanobody NbGFP-A647 was limited to the distal OS tips, and only a few rods had any Rho-GFP labeling in the proximal OS near the centrin-positive CC (Fig. 1 E) showing no detectable fluorescent signal in the IS (Fig. 1 F). This result changed when IS-enriched retinas from Rho-GFP/+ mice were used. In these retinas, NbGFP-A647 fluorescence was now observed in the rod CC, IS and ONL regions among the sparse remaining OSs in these sections (Fig. 2 E). Based on these results, the authors propose a model of rhodopsin post-Golgi vesicle trafficking where rhodopsin containing vesicles fuse with the IS plasma membrane and traffic to the periciliary ridge by 2D diffusion. 

Critique

This is an interesting manuscript employing state of the art imaging techniques to analyze localization of mouse rhodopsin or its hRho-GFP fusion proteins. Experiments are carefully carried out and statistically evaluated. This reviewer is unsure what it all means for GPCR trafficking in general. Authors firmly believe that fluorescence emitted by hRho-GFP represents the location of WT rhodopsin. The reviewer thinks that the hRho-GFP is a mutant, not identical to WT rhodopsin, and that its location in the IS PM differs from that of rhodopsin in WT animals. Further, OS peeling to generate IS-enriched retina may damage the IS, particularly the microtubule cytoskeleton and cytoplasmic dynein. This could be the reason why the authors cannot colocalize hRho-GFP and dynein. There is plenty of evidence that GPCR trafficking in neurons depends on molecular motors. Specifically rhodopsin strongly depends on dynein for delivery to the outer segment. 

In the hRho-GFP mouse lines, the C-terminal of human rhodopsin is fused to GFP and the mutant mimics the distribution of wild-type rhodopsin in hRho/+ heterozygotes (Chan et al., 2004). hRho/+ mice eventually degenerate, hRho-GFP acts as a recessive mutant. Presumably some aspect of the structure of the hRho-GFP is responsible for the degeneration. In hRho-GFP homozygotes, recessive retinitis pigmentosa is much accelerated suggesting that the hRho-GFP dimer is less stable than the hRho-GFP/mouse rhodopsin heterodimer, and mouse rhodopsin stabilizes hRho-GFP. Importantly, hRho-GFP is a mutant rhodopsin and foreign in the mouse retina. hRho-GFP/+ expresses both native mouse Rho and mutant hRho-GFP which form mouse Rho and mutant hRho-GFP dimers and presumably traffic as dimers. 

It is unclear why the fusion protein is unstable, or whether hRho-GFP's C-terminal CLS is partially blocked by the bulky GFP. It is unclear whether native mouse rhodopsin distribution in WT retina is identical to hRho-GFP/mouse rhodopsin distribution in the mutant. It is also unclear whether the amount of Rho-GFP is increased in IS-enriched rod IS, or only better visualized when OS are removed? 

The retinal peeling technique (Fig. 2A) to remove OS to generate IS-enriched retina samples is very interesting. Outer segments are successively removed in unfixed rods, where the retina slices were peeled 8 times using filter paper. The time needed for 8 peelings and the effects on the IS are unclear. Some CCs which are contiguous with the axoneme are likely pulled out, a procedure which must affect the microtubule cytoskeleton in the IS (compare 2F with 1F). The peeling technique could be called a mechanical photoreceptor degeneration technique with unknown consequences on IS metabolism. Removal of the basal body, functioning as microtubule organizing center, disrupts microtubule anchoring to the BB, a procedure that could inactivate cytoplasmic dynein with consequences on survival of rods. Inactivation of dynein could push vesicles to IS membrane after peeling is complete, increasing hRho-GFP at the IS membrane.

In the current form, the manuscript is not ready for publication in PlOS Biology.

Specific points:

Abstract: "Thus, our results collectively establish a model of rhodopsin trafficking through the inner segment plasma membrane as an essential subcellular pathway in mouse rod photoreceptors." As most of the rod IS results are based on GFP fluorescence, the results establish a model for hRho-GFP trafficking, not necessarily for mouse rhodopsin. 

Line 109: "Rho-GFP fusion knockin mouse." Chan et al. used human rhodopsin to generate the mouse line. hRho-GFP should be used instead of Rho-GFP. 

Line 137: "rhodopsin trafficking within the IS remains largely uncharacterized". The term rhodopsin transport yields nearly 2000 responses in PubMed. There are plenty of unknowns, but rhodopsin trafficking has been investigated to quite an extent. 

Line 175: "Rho protein in the IS was undetectable with immunofluorescence." But in 1A, there are traces of Rho in the IS with 1D4, 4D2. 

LIne 210: "Even when we doubled the amount of NbGFP-A647 used during labeling, the Rho-GFP-localized fluorescence was still predominantly found in distal OS tips of Rho-GFP/+ retinas with no detectable fluorescent signal in the IS (Fig. 1 F)." Therefore expression of hRho-GFP after OS peeling increases? This provides a hint that OS peeling damages the IS. 

Line 231: "IS-enriched rod examples had preserved CC structure that were typically attached to a remaining OS." Apparently, some CCs are pulled out together with OS which must affect the microtubule cytoskeleton in the IS (compare 2F with 1F).

Line 241: "Rhodopsin is localized at the mouse rod inner segment plasma membrane. " Rhodopsin should be replaced by hRho-GFP. 

Line 280-283:" In IS-enriched WT retinas that were immunolabeled with Rho-C-1D4 antibody, the endogenous Rho localization pattern matched the Rho-GFP IS pattern, including localization at the IS plasma membrane and CC membrane, in addition to prominent bright puncta labeling in the myoid IS." This could be due to hRho-GFP/mouse Rho heterodimer formation. 

Line 449: "Together these results indicate that vesicular Golgi and post-Golgi trafficking organelles are targeted to the IS PM in hRho-GFP/+ mouse rods." I added hRho-GFP/+ . 

Line 475. "Although we observed partial co-localization between Rho-GFP and DYNC1H1 with STORM, the overall abundant and diffuse pattern of IS DYNC1H1 is likely why the interaction strength with Rho molecules was not distinguishably higher than random." This maybe due to Dynein being in the inactive state, also called Phi state. If the microtubule cytoskeleton is damaged by OS peeling, Dynein stops interacting with cargo. The general assumption is that GPCR trafficking in neurons depends on molecular motors, see PMID: 35835604; specifically dynein, see Nemet et al. 2014, Dahl et al, 2021. 

Line 491: "suggest that after Golgi processing, new Rho proteins are trafficked within vesicles to the IS plasma membrane as a target/destination membrane." This is close to the standard hypothesis, that rhodopsin containing vesicles are unloaded at the IS PM near the periciliary ridge. In the author's model, unloading would happen anywhere at the IS PM. This still could be an artifact of the hRho-GFP mouse line. 

Line 498: "Furthermore, dense rod IS plasma membrane 499 localization of Rho was previously reported in pig retina (Röhlich et al., 1989), in mouse and 500 cow rods (Jan and Revel, 1974), and in developing rat rods (prior to OS formation) (Nir et al., 501 1984)." In older publications, this may depend on the quality of antibodies and fluorescence microscopes. 

Line 547: "Internal Rho that was sporadically localized in the ellipsoid IS may correspond to newly-synthesized Rho in the rough endoplasmic reticulum (ER)." The RER surrounds nuclei, this is where rhodopsin is biosynthesized. ER present in the ellipsoid must be smooth ER that is not involved in rhodopsin biosynthesis. 

Reviewer #3: The study by Haggerty et al. uses super-resolution microscopy to interrogate the subcellular localization of rhodopsin, the visual pigment in rod photoreceptors, in the mouse retina. The localization of rhodopsin and other known molecules is determined in rod inner segments specifically. Two super-resolution microscopic techniques are used: SIM and STORM. The authors present convincing images of high quality. The work is well-done, the manuscript easy to follow and the conclusions well-supported by the data. I think that globally, the study confirms what was already known about the presence of many of the molecules imaged here. However, there are also new findings, such as the colocalization of STX3 and rhodopsin in the IS plasma membrane. The authors did an excellent job at discussing their data with the published data. The results collectively establish the power of super-resolution microscopy to interrogate protein trafficking in photoreceptors. The data also supports a novel of rhodopsin trafficking through the inner segment plasma membrane.

---

## [Decision Letter · Decision Letter 2]

22 Nov 2023

Dear Dr Robichaux,

Thank you for your patience while we considered your revised manuscript entitled "Mapping rhodopsin trafficking in rod photoreceptors with quantitative super-resolution microscopy" for publication as a Research Article at PLOS Biology. This revised version of your manuscript has been evaluated by the PLOS Biology editors, the Academic Editor and two of the original reviewers.

Based on the reviews (attached below), we are likely to accept this manuscript for publication, provided you satisfactorily address the remaining minor points raised by Reviewer 2. Please also make sure to address the data and other policy-related requests stated below.

In addition, we would like you to consider a suggestion to improve the title:

"Super-resolution mapping in rod photoreceptors identifies rhodopsin trafficking through the inner segment plasma membrane as an essential subcellular pathway"

We expect to receive your revised manuscript within two weeks. 

*Published Peer Review History*

*Press*

Sincerely,

Ines

--

Ines Alvarez-Garcia, PhD

Senior Editor

PLOS Biology

ETHICS STATEMENT:

Thank you for providing an ethics statement. Please also include an approval number.

DATA POLICY: IMPORTANT - PLEASE READ CAREFULLY

Fig. 4C-E; Fig. 5D, E; Fig. 6A-G; Fig. 7H-K; Fig. 10F, M; Fig. S2C; Fig. S3A-D and Fig. S4A, B

***Please also make sure that the data deposited in Mendeley Data DO(I:10.17632/pv3ty3z68f.1, DOI: 10.17632/r4246hbxsw.1, and DOI: 10.17632/hmztjrwhtc.1.) is made publicly available at this stage.

Thank you for submitting a file with the original raw gels, however I have noted that two of them are mislabelled: Fig. 3 should be labelled Fig. 2 and Fig. 7 should be Fig. 8. Please correct this and submit the new file.

Reviewers' comments

Rev. 1: Jeannie Chen – note that his reviewer has signed his review

The authors have adequately addressed my comments.

Rev. 2: Wolfgang Baehr – note that this reviewer has signed his review

Minor comments:

Line 54 " Rho, which is synthesized in the rod photoreceptor inner segment (IS)." This is not correct. Rho is synthesized at the RER surrounding rod nuclei in the ONL. ONL is not part of the IS. Rho mRNA is exported from nuclei , translated by ER-associated ribosomes and incorporated into the ER membrane. From there, the trafficking pathway starts.

Line 229. Cilia should be connecting cilia

Line 541 "Our model for IS Rho trafficking in mouse rods (depicted in Figure 10)..". The model depicted in Figure 10 was deleted in R2. Please delete.

Line 981-1001: Yellow arrows are not mentioned in the legend of Figure 1.

Line 1239 Incomplete reference

---

## [Editor Report · Decision Letter 3]

10 Dec 2023

Dear Dr Robichaux,

Thank you for the submission of your revised Research Article entitled "Super-resolution mapping in rod photoreceptors identifies rhodopsin trafficking through the inner segment plasma membrane as an essential subcellular pathway" for publication in PLOS Biology. On behalf of my colleagues and the Academic Editor, Tom Baden, I am delighted to let you know that we can in principle accept your manuscript for publication, provided you address any remaining formatting and reporting issues. These will be detailed in an email you should receive within 2-3 business days from our colleagues in the journal operations team; no action is required from you until then. Please note that we will not be able to formally accept your manuscript and schedule it for publication until you have completed any requested changes.

PRESS

Sincerely, 

Ines

--

Ines Alvarez-Garcia, PhD

Senior Editor

PLOS Biology
